# Exciton transport in molecular organic semiconductors boosted by transient quantum delocalization

Samuele Giannini [1,5 ✉], Wei-Tao Peng[1], Lorenzo Cupellini [2], Daniele Padula [3], Antoine Carof [4] & Jochen Blumberger [1 ✉]

Designing molecular materials with very large exciton diffusion lengths would remove some of the intrinsic limitations of present-day organic optoelectronic devices. Yet, the nature of excitons in these materials is still not sufficiently well understood. Here we present Frenkel exciton surface hopping, an efficient method to propagate excitons through truly nano-scale materials by solving the time-dependent Schrödinger equation coupled to nuclear motion. We find a clear correlation between diffusion constant and quantum delocalization of the exciton. In materials featuring some of the highest diffusion lengths to date, e.g. the non-fullerene acceptor Y6, the exciton propagates via a transient delocalization mechanism, reminiscent to what was recently proposed for charge transport. Yet, the extent of delocalization is rather modest, even in Y6, and found to be limited by the relatively large exciton reorganization energy. On this basis we chart out a path for rationally improving exciton transport in organic optoelectronic materials.

[1] Department of Physics and Astronomy and Thomas Young Centre, University College London, WC1E 6BT London, UK. [2] Dipartimento di Chimica e Chimica Industriale, Universitá di Pisa, Via G. Moruzzi 13, 56124 Pisa, Italy. [3] Dipartimento di Biotecnologie, Chimica e Farmacia, Universitá di Siena, Via A. Moro 2, 53100 Siena, Italy. [4] Laboratoire de Physique et Chimie Théoriques, CNRS, UMR No. 7019, Université de Lorraine, BP 239, 54506 Vandoeuvre-lés-Nancy Cedex, France. [5] Present address: Laboratory for Chemistry of Novel Materials, University of Mons, Place du Parc 20, 7000 Mons, Belgium. ✉email: samuele.giannini.16@ucl.ac.uk; j.blumberger@ucl.ac.uk

When an organic semiconductor (OS) absorbs light, electrons can be excited from the valence to the conduction band forming Coulombically bound electron-hole pairs. These excitations were described by Frenkel as energetic quasiparticles, which he named excitons[1,2]. Frenkel excitons as we now call them can diffuse through the OS material or decay through various pathways, e.g., via fluorescence, intersystem crossing, phosphorescence, non-radiative transitions depending on the energetics of the system and its interactions. In several applications, in particular in organic photovoltaics (OPVs), OS materials with high diffusion lengths $L \propto \sqrt{D\tau}$ (where $D$ and $\tau$ are the exciton diffusion constant and lifetime, respectively) are highly desirable. Larger diffusion lengths mean larger domain sizes of the photoactive layer can be used, which would simplify device design and improve power conversion efficiency[3]. While classical molecular OSs, suitable for OPV applications, have diffusion lengths of no more than 5–20 nm[4], several examples of new materials have recently been reported, termed non-fullerene acceptors (NFAs), that support longer range exciton transport, 20–47 nm[5]. The reason for this and the relations between molecular properties/packing and exciton transport are still not well understood. Establishing better computational approaches would help us to understand the conditions in which exciton diffusion lengths are very large and beyond the values of today's best molecular materials.

In an extended system such as a molecular crystal, aggregate, or polymer, excitons can be shared between molecular sites depending on the strength of the electronic interactions between the sites. As a consequence, the excited state spectrum of the material will differ from that of the isolated molecule[6,7]. Recent theoretical and numerical advances in the related area of charge transport have provided strong evidence that charge carrier diffusion (i.e., mobility) in molecular materials is strongly correlated with the extent of charge carrier delocalization[8–10]. One can expect that the same is true for excitons. Hence, computational modeling of materials with high exciton diffusivity need to take into account and be able to predict the extent of exciton delocalization in the material. This requires the development of efficient approaches that can be applied to system sizes large enough to fully accommodate delocalized excitons. Moreover, the method needs to take into account thermal vibrations that couple to the electronic interactions between the molecules (termed diagonal and off-diagonal exciton-phonon coupling) as well as the so-called "back reaction", namely the effect of the electronic interactions on the nuclear motion. Standard exciton hopping theories based on Fermi's Golden Rule, e.g., Marcus-Levich-Jortner, assume localized excitons and are thus expected to be of limited use for the modeling of materials with high exciton diffusion constants[11,12]. Accurate quantum dynamics approaches like multi-layer multiconfigurational time-dependent Hartree (ML-MCDTH) are arguably the gold standard for such problems—however, they are typically limited to relatively small 1D model systems of a few hundred vibrational degrees of freedom. While they are excellent for benchmarking more approximate methods, they become computationally expensive, if not prohibitive, for truly nanoscale systems[13].

In this work, we introduce Frenkel exciton surface hopping (FE-SH), a computationally efficient atomistic non-adiabatic molecular dynamics method that strikes an optimal balance between predictive power and computational feasibility and that addresses the desirable criteria outlined above. In this method, the excitonic wavefunction is represented as linear combinations of molecularly localized or quasi-diabatic Frenkel excitons interacting via excitonic couplings. This wavefunction is propagated across the material by solving the time-dependent Schrödinger equation coupled to the finite temperature motion of the nuclei, and feedback from the electronic to the nuclear degrees of freedom are incorporated by stochastic hops between the excitonic potential energy surfaces (see Methods for details). A defining feature of FE-SH is that electronic structure calculations are only carried out for parametrization of the Frenkel exciton Hamiltonian matrix elements, i.e. site energies, excitonic couplings, and their thermal fluctuations, but not during FE-SH dynamics. In this respect, our method differs from the approach by Sisto et al.[14,15], where surface hopping coupled with an ab initio derived "on-the-fly" exciton Hamiltonian was developed to simulate exciton transfer along a few tens of chromophores (~3000 atoms). FE-SH allows us to carry out non-adiabatic MD simulations of exciton transport on yet larger, truly nanoscale systems (>10 nm, ~$10^3$ molecules, ~$10^5$ atoms) on the ps time scale.

Conceptually, the FE-SH method is similar to our previously presented fragment-orbital-based surface hopping method (FOB-SH) for charge transport[9,16–18]. An important difference with respect to FOB-SH is that the short-ranged (exponentially decaying) electronic couplings and nuclear derivatives in FOB-SH are now replaced by the long-ranged excitonic couplings in FE-SH. Taking into account the long-range nature of the latter has been shown to be important in a recent coarse-grained dynamics study of Frenkel excitons in polymer fibers[19,20]. While our fully atomistic scheme can be relatively easily extended to include relaxation processes of Frenkel excitons such as exciton dissociation and recombination, in this work we place the focus solely on singlet exciton transport.

Here we apply FE-SH to five different molecular organic crystals as shown in Fig. 1, anthracene (ANT)[21], α-sexithiophene (a6T)[22], perylenetetracarboxylic diimide (PDI)[23], dicyanovinyl-capped $S,N$-heteropentacene (DCVSN5)[24], and Y6[25]. These systems are chosen to cover a wide range of diffusive regimes as well as for their relevance in organic optoelectronics. ANT has been very well studied experimentally[26–28] and computationally[29,30] providing a good test case for our new FE-SH methodology. Polycrystalline a6T was recently shown to support exciton transport and dissociation without the presence of an acceptor material making it a good candidate for the development of OPV homojunctions[31]. It is also a good model system for thiophene-based polymeric light absorbers[19] and electron donors in OPV heterojunctions. DCVSN5 is of interest because of the high exciton diffusion constant predicted for this material[29] and because it was used as an electron donor in fullerene-based heterojunctions with moderate power conversion efficiency (about 6.5%)[29,32]. Finally, we study two non-fullerene acceptor materials[33], PDI as a representative of the chemically highly tunable class of perylene diamides[34], as well as Y6[25,35]. The latter has an intriguing chemical structure composed of alternating acceptor and donor moieties in the form A-DAD-A and it was used as the acceptor material in OPV devices with record-breaking power conversion efficiencies >18%[35,36].

## Results

The FE-SH algorithm, presented in detail in the Methods section, implements the excitonic time-dependent Schrödinger equation (Eq. (4)) using the time-dependent Frenkel exciton Hamiltonian (Eq. (2)). In the following, we investigate two key assumptions of our approach before presenting applications of FE-SH to molecular crystals. First, we establish that the Frenkel exciton Hamiltonian provides a suitable description of the electronic transitions in the systems studied. Second, we show that the total excitonic couplings between these states is well approximated by

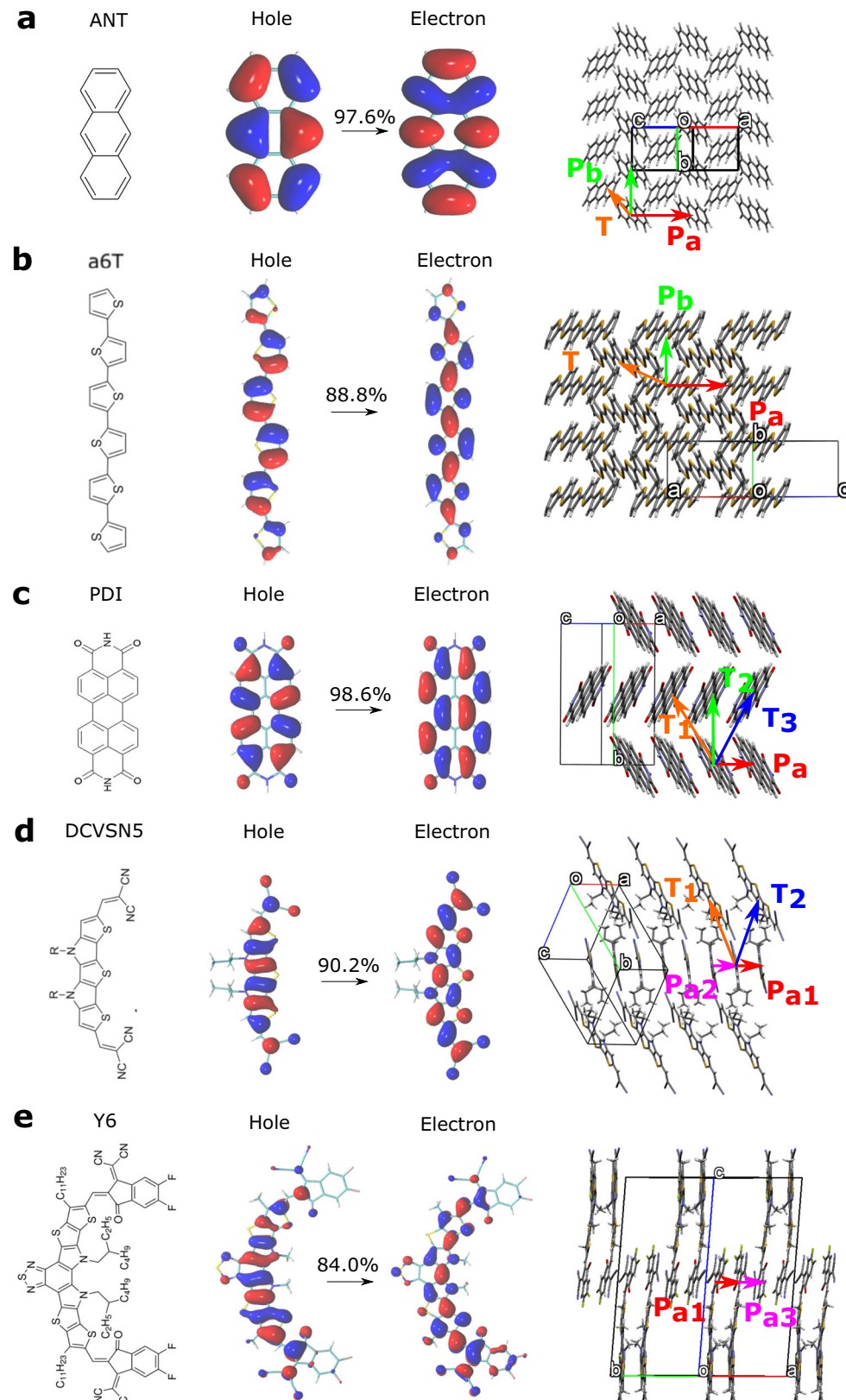

**Fig. 1 Organic molecular crystals investigated in this work.** The chemical structures of the single molecules (left panels), natural transition orbitals (NTOs) for the first singlet excited state (middle panels), and the experimental crystal structure (right panels) are shown for anthracene (ANT, **a**), α-sexithiophene (a6T, **b**), perylenetetracarboxylic diimide (PDI, **c**), dicyanovinyl-capped S,N-heteropentacene (DCVSN5, **d**), and (Y6, **e**). Directions of strongest excitonic couplings are indicated in the crystal structures and labeled according to their crystallographic direction (e.g., $P_a$, $P_b$, $T$, see also Table 1). In Y6, $P_{a2}$ is eclipsed by $P_{a1}$. The unit cell axes $a$, $b$, $c$ are shown in red, green, and blue. The percentages indicated denote the contribution to the S1 singlet excited states due to the NTOs shown. The NTOs labeled "Hole" and "Electron" resemble very closely the HOMO and LUMO of each molecule.

the Coulomb contribution only, similarly as in biological photosystems[37,38]. The latter observation is particularly important because it allows us to apply to method to truly nanoscale systems with little loss in accuracy and at moderate computational cost (details on the calculations presented below are given in Methods).

**S0 → S1 electronic transitions**. We consider at first the S1 excited state of the isolated molecules and analyse the natural transition orbitals (NTOs), shown in Fig. 1. The NTOs offer a useful way of visualizing which orbitals give the largest contribution to a given single-particle excitation. For all systems, we can see that the NTOs with the largest contributions to the S0 → S1 electronic transition resemble very closely the highest occupied molecular orbital (HOMO) and lowest unoccupied molecular orbital (LUMO) of the single molecules. Moreover, the second-lowest singlet excited state, S2, is at least ~0.5 eV above S1 in all systems at the level of theory used in this work (values given in Supplementary Table 1). This implies that the low-energy tail of the band formed by the lowest intra-molecular excitation in the crystal is well described by a linear combination of the intramolecular S1 states, and the latter, to a good approximation, by HOMO-LUMO transitions, attesting to the validity of the Frenkel exciton picture. Potentially lower-lying excited states, e.g., intermolecular charge transfer states may exist, but they are not considered here as the focus is placed solely on the transport dynamics of Frenkel exciton states. Charge transfer states could be included by the expansion of the state space of our electronic Hamiltonian, which is, however, beyond the scope of the present work[39].

**Excitonic couplings**. Next, we turn to the calculation of excitonic couplings between the Frenkel states $k$ and $l$, $V_{kl}$ (in the following we skip the molecular indices $k, l$ for convenience). $V$ is typically written as a sum of a long-range or Coulombic part, $V^{Coulomb}$, and a short-range part, $V^{short}$, accounting for exchange, overlap, and electronic polarization contributions, $V = V^{Coulomb} + V^{short}$[40]. In applications of FE-SH to nanoscale systems a very large number of excitonic couplings need to be calculated on-the-fly (typically >10$^9$ per trajectory) which mandates the use of an ultrafast coupling estimator. We do this by approximating $V$ by the long-range part $V^{Coulomb}$ (Eq. (11)) and the latter by $V^{TrESP}$ (Eq. (12)) using frozen transition electrostatic potential (TrESP) charges in place of transition densities, $V \approx V^{Coulomb} \approx V^{TrESP}$[37,41]. These two approximations introduce little error as shown in Table 1 and Fig. 2. We find that $V^{Coulomb}$ obtained from the S1 excitation of the isolated molecules is very close to the total coupling $V$ (Fig. 2a, mean relative unsigned error (MRUE) ≈ 6.6%). $V$ accounts for all short- and long-range effects and was obtained by diabatization of the electronic excitations of molecular dimers using the multi-state fragment excitation energy difference-fragment charge difference method (MS-FED-FCD) in combination with TDDFT[42–44]. MS-FED-FCD, in turn, agrees well with available excitonic coupling from spin component scaled approximate coupled-cluster theory (SCS-CC2) (27.9 and 25.6 meV for $P_b$ pair of anthracene[11], respectively). Thus, our results indicate that the short-range coupling $V^{short}$ is of little importance even for the closely packed organic molecular solids studied here and can be neglected to a good approximation. A detailed discussion on this conclusion is given in Supplementary Note 3 and the importance of adopting a multi-state diabatization procedure to recover accurate excitonic couplings between (maximally) localized Frenkel exciton states is shown in Supplementary Figs. 1, 2 and discussed in Methods.

In addition to the above, we find that $V^{Coulomb}$ is well approximated by $V^{TrESP}$ (Fig. 2b, MRUE = 7.0%). Moreover, the thermal fluctuations of $V^{Coulomb}$ along molecular dynamics trajectories with the transition densities updated at every MD step are very well reproduced by $V^{TrESP}$ with frozen transition charges (see Supplementary Figs. 3, 4). These results are very significant because the calculation of $V^{TrESP}$ is orders of magnitude faster than for $V^{Coulomb}$ allowing us to apply FE-SH to very large, nanoscale systems. Moreover, it permits including excitonic couplings beyond nearest-neighbor pairs, which can be significant due to their long-range nature ($V \propto r^{-3}$) and important for an accurate estimation of exciton diffusion constants in certain systems, as shown previously[19,20]. Interestingly, the commonly used point dipole approximation (PDA) to $V^{Coulomb}$, $V^{PDA}$ (Eq. (8) in Supplementary Method 1), is found to break down for all molecules except ANT suggesting caution in the application of (dipole-based) Förster theory to these materials (see Supplementary Fig. 5). Further discussion of excitonic coupling calculations is given in Supplementary Method 1.

**Exciton transport mechanism from FE-SH non-adiabatic molecular dynamics**. The exciton wavefunction, $\Psi(t)$ (Eq. 3), is propagated in time across the $a − b$ planes of ANT, a6T, PDI, and along rod-like nanocrystals composed of 5 and 4 columnar stacks for DCVSN5 and Y6, respectively, oriented along the $a$-direction (each system contains about 300 molecules). Notably, our FE-SH methodology obeys a number of desirable properties including energy conservation, internal consistency, and detailed balance in the limit of long simulation times[17,18,45]. We refer to the Methods section (and Supplementary Note 4) for full simulation details of the systems investigated.

In the following, we focus on the transport scenario in two systems that exhibit the lowest (ANT) and the highest (DCVSN5) exciton delocalization of the five systems investigated. For ANT we find that the exciton remains localized on a single molecule for most of the time, as depicted by a value of the inverse participation ratio (IPR, Eq. 8) that is close to unity when averaged over time, ⟨IPR⟩ ≈ 1 (Fig. 3a, bottom panel). Thermal fluctuations occasionally lead to short-lived electronic excitations within the excitonic band (peaks in Fig. 3a middle and upper panels) accompanied by delocalization over a single nearest neighbor (i.e., increase in IPR to values of up to two, bottom panel in Fig. 3a). This is followed by de-excitation and concomitant re-localization of the exciton to either the neighboring molecule, in which case the exciton becomes displaced as shown in Fig. 3c, or back to the original molecule. Short-lived, more extended delocalizations of up to four molecules are also observed, but those are much rarer than the two-site delocalization events described above. Hence, our simulations show that ANT is a good example of the nearest-neighbor hopping of a fully localized exciton.

This is no longer the case for the more diffusive DCVSN5 system. The exciton is delocalized over two molecules on average, ⟨IPR⟩ ≈ 2 (Fig. 3b, bottom panel). Thermal excitations to higher-lying states within the excitonic band are more frequent than for ANT (Fig. 3b, upper and middle panels) resulting in short-lived expansions of the exciton over up to about 15 molecules. We define events where IPR($t$) > ⟨IPR⟩ + 1 as "transient delocalization" to distinguish them from local or short-range exciton transfer events where, during the transition, IPR = ⟨IPR⟩ + 1, but not larger than that. This would describe, for instance, hopping of a fully localized exciton to one of its nearest neighbors, where IPR changes from 1 to 2 in the transition state and back to 1 after the transition, or a shift of a delocalized polaron by one molecular unit. The transient

**Table 1 Computed parameters for exciton transport in the molecular OSs shown in Fig. 1. All values in meV (except distances in Å)[a].**

| | | Dist. [b] | $V_{kl}$ [c,d] | $V_{kl}^{Coulomb}$ [c] | $V_{kl}^{TrESP}$ [e] | $\langle V_{kl}^{TrESP}\rangle$ [f] | $\sigma_{kl}^{TrESP}$ [g] | $\lambda^{XT}$ [h] |
|---|---|---|---|---|---|---|---|---|
| ANT | $P_b$ | 6.04 | −27.9 | −25.9 | −26.3 | −29.5 | 3.2 | 561 |
| | $T$ | 5.24 | 6.4 | 4.5 | 4.5 | 4.3 | 5.3 | |
| | $P_a$ | 8.56 | 4.4 | 4.3 | 4.2 | 4.5 | 0.8 | |
| a6T | $P_b$ | 5.68 | 91.9 | 88.0 | 86.9 | 86.5 | 2.7 | 558 |
| | $T$ | 5.38 | −101.2 | −96.1 | −95.4 | −93.6 | 3.4 | |
| | $P_a$ | 9.14 | 42.0 | 41.8 | 41.6 | 40.6 | 2.0 | |
| PDI | $P_a$ | 4.87 | 100.1 | 85.7 | 85.5 | 105.7 | 8.4 | 390 |
| | $T_1$ | 9.47 | −32.0 | −31.9 | −31.8 | −28.7 | 1.4 | |
| | $T_2$ | 9.40 | −17.9 | −17.0 | −17.1 | −12.9 | 2.8 | |
| DCVSN5 | $P_{a1}$ | 3.64 | −137.6 | −135.6 | −135.0 | −131.8 | 6.1 | 320 |
| | $P_{a2}$ | 4.46 | −151.9 | −141.6 | −142.0 | −139.2 | 7.9 | |
| | $T_1$ | 14.20 | 24.1 | 24.4 | 24.3 | 22.4 | 2.9 | |
| Y6 | $P_{a1}$ | 9.41 | −86.4 | −73.7 | −63.1 | −62.2 | 3.3 | 250 |
| | $P_{a2}$ | 15.35 | −61.3 | −55.2 | −63.3 | −61.9 | 4.5 | |
| | $P_{a3}$ | 18.34 | 52.6 | 47.4 | 47.5 | 39.8 | 5.0 | |

[a]Excitonic couplings (meV) are computed for a few of the closest crystal pairs, as shown in Fig. 1. The subscripts approximately indicate the crystallographic direction along which dimers are oriented. The basis set is fixed to 6-31g(d,p), CAM-B3LYP functional is used for a6T, PDI and Y6, ωB97X-D functional for ANT and DCVSN5. The choice of the DFT functional is discussed in Methods and more extensively in Supplementary Note 1 and Supplementary Method 1.
[b]Distance (in Å) for the pairs indicated.
[c]The sign of $V_{kl}$ and $V_{kl}^{Coulomb}$ (Eq. (11)) is adjusted according to the sign obtained from $V_{kl}^{TrESP}$ (Eq. (12)), where a consistent phase of the transition density on all the molecules is given by construction. A negative coupling does not necessarily mean J-aggregation in this context as the coupling sign depends on the relative orientation of the interacting transition dipoles (see also Supplementary Fig. 6).
[d]$V_{kl}$ couplings are evaluated using the MS-FED-FCD diabatization approach (using the first 20 (singlet) excited states for the diabatization procedure, see main text and Supplementary Method 1).
[e]TrESP are parametrized as described in Methods and using the level of the theory mentioned above.
[f]Mean excitonic couplings are averaged over values extracted from all the pairs (in the given direction) for all the time snapshots of ten FE-SH trajectories of at least 1 ps.
[g]Fluctuations of electronic couplings from the aforementioned trajectories, $\sigma_{kl}^{TrESP}=\sqrt{\langle(H_{kl}^{TrESP}-\langle H_{kl}^{TrESP}\rangle)^2\rangle}$.
[h]Reorganization energies (meV) corresponding to the S1 state, $\lambda^{XT}$, are computed according to Eq. (10) using the same level of theory as for coupling calculations.

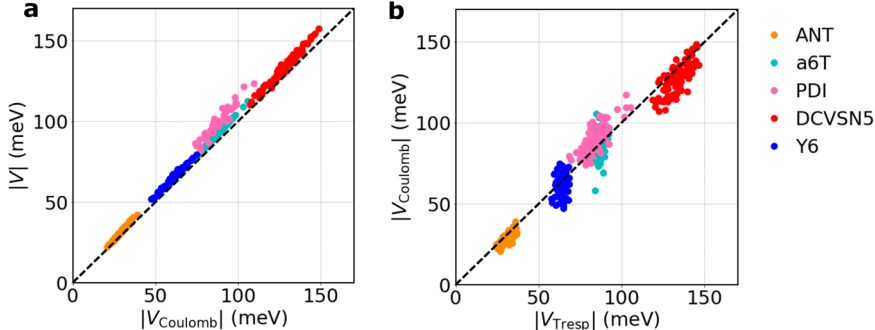

**Fig. 2 Accuracy of fast excitonic coupling estimators. a** Correlation between the reference total excitonic coupling $V$ (in absolute value), obtained using MS-FED-FCD (Eq. (7) in Supplementary Method 1), and $V^{Coulomb}$ (Eq. (11)) for neighboring molecules along the π-stacking direction ($P_b$ of ANT and a6T and $P_a$, $P_{a1}$, $P_{a2}$ for PDI, DCVSN5, and Y6 respectively). Excitonic couplings were calculated for configurations taken from MD trajectories generated for supercells of the crystals at 300 K. $V$, couplings correspond to the off-diagonal block $\mathbb{H}'''_{XT1,XT2}$ in MS-FED-FCD related to the lowest localized excited states in molecules $k$ and $l$, respectively and the diabatization was performed using the first 20 adiabatic states, which proved to be sufficient for convergence (see Supplementary Note 3). **b** Correlation between $V^{Coulomb}$ (Eq. (11)) and $V_{TrESP}$ (Eq. (12)). Mean relative unsigned error (%) is defined as MRUE $=(\sum_n(|y_{calc}-y_{ref}|/y_{ref}))/n$. In both panels the MRUE is ca. 7%. This small deviation found for the systems investigated justifies the use of $V_{TrESP}$ in FE-SH simulations.

delocalization events account for 8.4% of the full IPR distribution (Fig. 4b, section shaded in yellow). Averaging over all trajectories, we find that they occur about every ≈90 fs and typically last less than 8 fs. Importantly, some of these events facilitate coherent exciton transfer to molecules up to 3–4 lattice spacings apart (≈20–30 Å) permitting much larger spatial displacements than in ANT, see Fig. 3d. In other words, in DCVSN5 exciton transfer is no longer bound to nearest neighbor hops, as was the case for ANT, but can occur across several molecules at a time, i.e., coherently, resulting in a boost of the diffusion constants, as quantified further below. Y6 exhibits a similar transient delocalization mechanism and the transport scenario in the other materials studied is between the two limiting cases described.

**Exciton diffusion constants.** The mean-squared displacement (MSD) of the exciton wavefunction averaged over about 600 FE-SH trajectories is shown in Supplementary Fig. 8a–e for each system. We find that the MSD increases rapidly at short times (<100 fs) followed by a more shallow and approximately linear increase at longer times. The initial rapid increase is due to electronic relaxation of the initially fully localized exciton (which is a linear combination of eigenstates of the excitonic Hamiltonian, Eq. (2)) to states at the low energy tail of the excitonic band (see Methods for details). While the short-time relaxation dynamics within the excitonic band formed by the Frenkel states depends on the choice of the initial exciton wavefunction, the long-time diffusive dynamics is the same for different initial

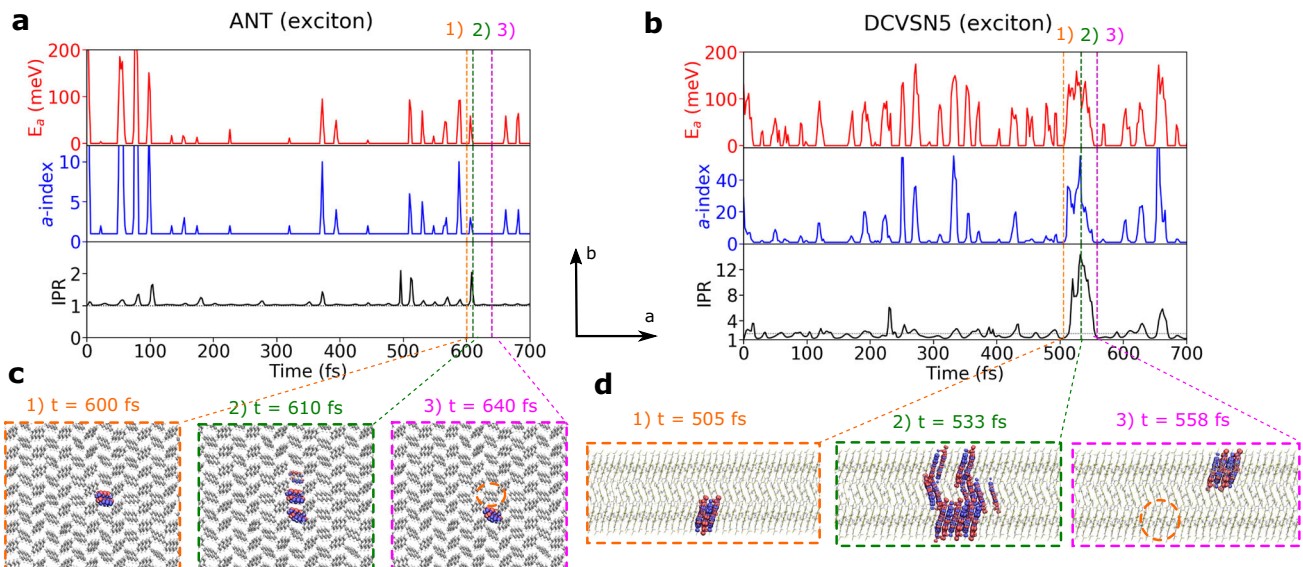

**Fig. 3 Hopping and transient delocalization mechanisms for exciton transport.** Panels **a**, **b** show time series along a single representative FE-SH trajectory for ANT and DCVSN5, respectively. Top panels: energy of the active potential energy surface on which the nuclear dynamics is run in FE-SH simulations, $E_a$ (red lines), referred to the ground state. Middle panels: index of the active exciton band state, $a$ (blue lines, index $a = 0$ corresponds to the ground state which is the bottom of the excitonic band). Bottom panels: IPR($t$), Eq. (8), of the excitonic wavefunction $\Psi(t)$, Eq. (3), (black lines) and average IPR of $\Psi(t)$ over the swarm of FE-SH trajectories (horizontal dashed gray lines). Note the correlation between the intra-band excitations, i.e., eigenstate index $a$, excitation energy, and IPR($t$) of $\Psi(t)$. Panel **c** depicts a typical nearest-neighbor hopping event of the exciton in ANT and panel **d** a representative transient delocalization event of the exciton in DCVSN5. Notice the much larger spatial displacement from the initial position (indicated with orange circles) in the latter case. The excitonic wavefunctions in panels (**c**, **d**), $\Psi(t)$, Eq. (3), are represented by isosurfaces of the transition density on each molecule scaled by the expansion coefficient $u_l$. The transition density is approximated, for visualization purposes, by the conjugated product of HOMO and LUMO orbitals.

states[46,47]. This is because FE-SH fulfils detailed balance in the long-time limit to a very good approximation[17,45]. This essential condition ensures that after initial relaxation, independently of the initial starting point, the populations of the excitonic band states reach the Boltzmann populations at long times. See Methods for details and Supplementary Fig. 11 where simulations are initialized from electronic eigenstates. The linear increase of the MSD at longer times is characteristic of Einstein diffusion and this is the regime for which we extract the diffusion tensor Eq. (6) (dashed black lines in Supplementary Fig. 8a–e). The diffusion constants along different crystallographic directions are well converged with regard to system size for the herringbone planes and rod-like nano-crystal shapes used in this work for the systems investigated (see Supplementary Fig. 9a). In Fig. 4a, we show that the diffusion constants span two orders of magnitude in the materials studied, they are highest in the direction of highest excitonic coupling and exhibit a clear correlation with the average exciton delocalization (⟨IPR⟩). The latter quantity is also well converged for the supercells considered in this work (see Supplementary Fig. 9b).

The magnitude and anisotropy of the computed diffusion constants are in good agreement with available experimental data. For ANT we obtain $D_b = 3.3 \times 10^{-3}$ cm$^2$ s$^{-1}$ and $D_a = 0.77 \times 10^{-3}$ cm$^2$ s$^{-1}$ compared with $D_b = 5.0 \times 10^{-3}$ cm$^2$ s$^{-1}$ and $D_a = 1.8 \times 10^{-3}$ cm$^2$ s$^{-1}$ estimated using the experimentally measured diffusion lengths, $L_{exp}$ (see Table 2). For this system, both diffusion constants obtained from FE-SH are slightly underestimated compared to experiments, similarly to previous calculations by Elstner and co-workers who employed Boltzmann-corrected Ehrenfest dynamics for this system[30]. A possible reason is the neglect of nuclear tunneling effects in our simulations, which can become significant for large activation barriers for the excitation energy transfer process, like in ANT. Another possible explanation is the absence of mixing between Frenkel exciton and charge-transfer states

(FE-CT) in present simulations, which, for some acenes, is known to enhance singlet exciton diffusion as a consequence of the larger exciton dispersion[48]. For an extended discussion of other computational methods applied to this system, including rate theory, see Supplementary Note 6 and Supplementary Table 5.

A similarly good agreement with experiments is obtained at the higher end of diffusion constants, $D = 150 \times 10^{-3}$ cm$^2$ s$^{-1}$ for Y6 from FE-SH compared with an estimated $D = 54 \times 10^{-3}$ cm$^2$ s$^{-1}$ from experimentally determined diffusion lengths[5]. For this system, measurements were carried out for thin-film samples rather than single crystals. The static disorder present in the thin films could explain the somewhat smaller diffusion constant in the experiment. We also mention that wavefunction delocalization and diffusion constant of Y6 from the present simulations, although in good agreement with experiments, might constitute a lower limit as a more extended structural model in $b$ and $c$ directions might slightly increase the IPR and exciton diffusion in this system. The fact that the diffusion constant in Y6 is at least one order of magnitude larger than in ANT supports the picture from FE-SH simulation of a diffusion mechanism that is qualitatively different and more efficient than nearest-neighbor hopping, similarly to what was found in Fig. 3b for DCVSN5. For the latter and other materials, to the best of our knowledge, no experimental estimates for exciton diffusion constants are available but qualitative comparisons to exciton diffusion lengths can be made which are again rather favorable, see Supplementary Note 6 for a discussion.

**Impact of transient delocalization on diffusion.** Transient delocalization events, by which the exciton wavefunction delocalizes over several molecules at a time (as seen in Fig. 3b), have a strong impact on the diffusion constant. We quantified this effect for DCVSN5 by calculating a modified MSD of the exciton wavefunction, $\Psi(t)$, where all displacements that occur via

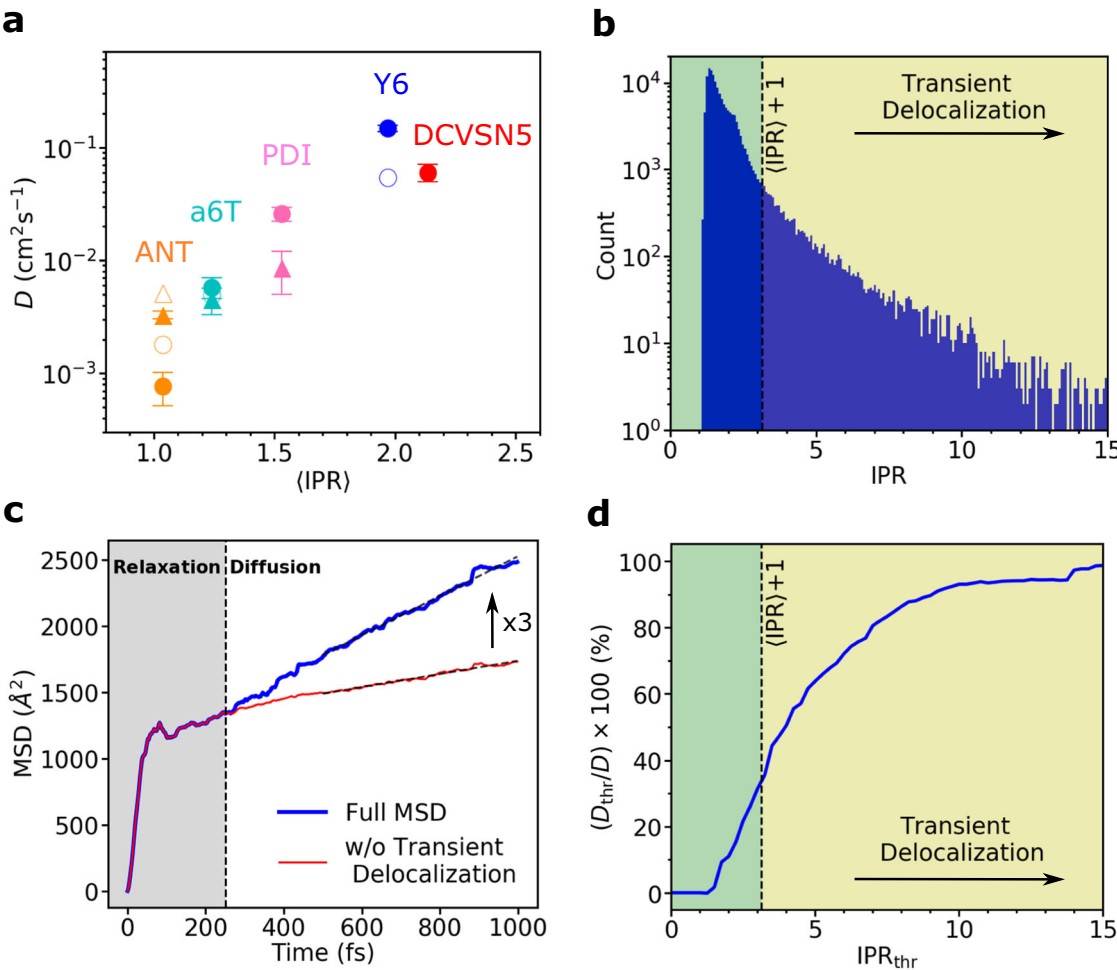

**Fig. 4 Exciton diffusion constant and impact of transient delocalization. a** Exciton diffusion constant ($D$) as a function of the average IPR, ($\langle IPR \rangle$), for the systems investigated (see Fig. 1 for their chemical and crystal structures). The diffusion constants along the crystallographic directions $a$ and $b$ obtained from FE-SH simulations are represented by filled circles and filled triangles pointing upwards, respectively. The average of $D$ and $\langle IPR \rangle$ values over different system sizes is plotted. Numerical values for specific system sizes are shown in Supplementary Fig. 9. Error bars indicate the corresponding standard deviations. Diffusion constants obtained from experimental data, as described in Table 2, are indicated by empty symbols and are placed at the computed IPR value since experimental estimates for exciton delocalization are not available. The crystallographic direction for the experimental diffusion constant for Y6 and a6T was not reported. **b** IPR distribution obtained from 600 FE-SH trajectories for a rod-like DCVSN5 nano-crystal comprised of 300 molecules. **c** The full MSD of the exciton wavefunction for DCVSN5 is shown with a blue line and is calculated according to Eq. (7). The MSD obtained when transient delocalization events are excluded is shown with a thin red line. In the latter case, the IPR threshold was set to $IPR_{thr} = \langle IPR \rangle + 1$. Notably, a threefold increase in the diffusion constant is obtained when transient delocalization events are retained. **d** Accumulated percentage contribution of wavefunction delocalization events to $D$ (i.e., $(D_{thr}/D) \times 100$), plotted as a function of the IPR threshold. The green region indicates the contribution of nearest-neighbor hopping to $D$, whereas the yellow region indicates the contribution due to transient delocalization.

transient wavefunction expansion beyond a given IPR threshold ($IPR_{thr}$) are discarded. The MSD obtained for a threshold $IPR_{thr} = \langle IPR \rangle + 1 \approx 3$, which includes only nearest neighbor shifts of the ~2-site delocalized polaron in DCVSN5, is shown in Fig. 4c (red line). In this case, the slope and therefore the diffusion constant is a factor of 3 smaller than for the full MSD (blue line) that includes transient delocalization events (defined as $IPR > \langle IPR \rangle + 1$, see above) showing that the latter boost diffusion significantly (three-fold). To generalize our results, in Fig. 4d we show the diffusion constant, $D_{thr}$, corresponding to an IPR threshold, $IPR_{thr}$, as a fraction of the total diffusion constant $D$. In the limit of large $IPR_{thr}$, i.e., all transitions included irrespective of the extent of delocalization, $D_{thr} \rightarrow D$. We find a steady increase in the diffusion constant with $IPR_{thr}$. Even at $IPR_{thr} = 2\langle IPR \rangle \approx 4$, only about half of the diffusion constant is accounted for. More extended delocalization events with $IPR > 2\langle IPR \rangle$, which happen only about 5% of the time according to Fig. 4b, contribute the

remaining 50% to the diffusion constant. This analysis clearly shows the major impact of transient delocalization events on exciton diffusion.

## Discussion

Exciton diffusion and charge carrier transport are essential processes underpinning the function of organic optoelectronic devices. Yet, how does the diffusivity and delocalization of excitons compare to the ones of charge carriers? In the following, we investigate this question by comparing the diffusion constants and IPRs for exciton transport with the values we obtained previously for charge transport using a similar surface hopping non-adiabatic molecular dynamics approach (termed FOB-SH)[8,9,49]. The results can only be indicative, as we compare exciton and charge diffusion constants for different sets of molecules, except in case of ANT where both values are available. To this end, we

**Table 2 Wavefunction delocalization ($\langle$IPR$\rangle$), exciton diffusion constants ($D$), and exciton diffusion lengths ($L$).**

| Systems | Dir. | FE-SH | | | Experiment | | |
|---|---|---|---|---|---|---|---|
| | | $\langle$IPR$\rangle$ | $D$ ($10^{-3}$ cm$^2$ s$^{-1}$) [a] | $L$ (nm) [b] | $\tau_{\text{exp}}$ (s) | $D_{\text{exp}}$ ($10^{-3}$ cm$^2$ s$^{-1}$) [c] | $L_{\text{exp}}$ (nm) |
| ANT | $a$ | 1.0 | 0.8 ± 0.3 | 39 ± 6 | $1.0 \times 10^{-8}$ [d] | 1.8 [e] | 60 [f] |
| | $b$ | | 3.3 ± 0.8 | 81 ± 9 | | 5.0 [e] | 100 [f] |
| a6T | $a$ | 1.2 | 6 ± 1 | 46 ± 5 | $1.8 \times 10^{-9}$ [g] | 4.9 [h] | 60 [i] |
| | $b$ | | 4.5 ± 0.3 | 40 ± 1 | | | |
| PDI | $a$ | 1.5 | 26 ± 4 | - | - | - | - |
| | $b$ | | 9 ± 3 | - | - | - | - |
| DCVSN5 | $a$ | 2.1 | 60 ± 11 | - | - | - | - |
| Y6 | $a$ | 2.0 | 150 ± 7 | 87 ± 2 | $2.5 \times 10^{-10}$ [j] | 54 [k] | 37 [j] (90 [l]) |

[a] In FE-SH $D$ is directly computed using Eq. (6). The diffusion constants were averaged over FE-SH simulations carried out for different system sizes, as reported in Supplementary Fig. 9. Error bars indicate the corresponding standard deviations.
[b] $L = \sqrt{2D\tau_{\text{exp}}}$.
[c] Usually the diffusion constant, $D_{\text{exp}}$, is not directly measured in experiments. In this case, we estimated $D_{\text{exp}}$ using the experimentally observed $L_{\text{exp}}$.
[d] Taken from ref. [26].
[e] Estimated considering $D_{\text{exp}} = L_{\text{exp}}^2 / 2\tau_{\text{exp}}$ assuming that the transport occurs in different 1D directions as done in ref. [29].
[f] Taken from ref. [27,28].
[g] Taken from ref. [72].
[h] Estimated assuming that a6T forms a 2D thin-film and the diffusion occurs isotropically within the herringbone plane, so that $D_{\text{exp}} = L_{\text{exp}}^2 / 4\tau_{\text{exp}}$.
[i] Taken from ref. [73]. Note that $L_{\text{exp}}$ in this case refers to a6T thin-film morphology and it should be taken as indicative only when comparing with the computed value (see Supplementary Note 6 for a discussion).
[j] Taken from ref. [5]. $L_{\text{exp}}$ in this case refers to Y6 thin-film morphology and it should be taken again as indicative only.
[k] Taken from ref. [5], where a 3D model was used to estimate this value.
[l] Estimated assuming isotropic exciton transport in 3D and taking $D_{\text{exp}}$ and $\tau_{\text{exp}}$ from ref. [5]. In this case $L_{\text{exp}} = \sqrt{6D_{\text{exp}}\tau_{\text{exp}}}$.

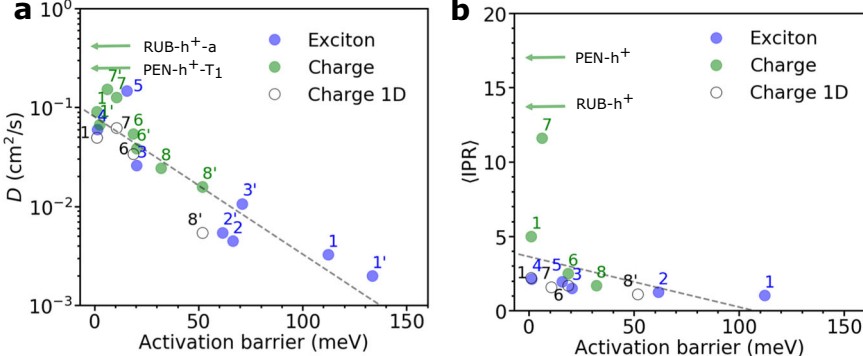

**Fig. 5 Comparison charge vs exciton transport.** Correlation between **a** the diffusion constant ($D$, Eq. (6)) and **b** the time average inverse participation ratio (IPR, Eq. (8)) of charges and excitons against the activation barrier (Eq. (1)) for different systems, respectively. Data from present exciton transport simulations are depicted in blue (ANT (1), a6T (2), PDI (3), DCVSN5 (4), and Y6 (5)). Data for charge transport simulations for the 2D conductive layers of OSs are taken from ref. [49] and depicted in green (ANT (1), naphthalene (6), perylene (7), pMSB (8), rubrene (RUB), and pentacene (PEN)). For 2D simulations, two data points are shown for diffusion constant and activation barrier, one for the direction of highest coupling (unprimed number) and one for the direction of second highest coupling (primed number). PEN-h$^+$-$T_1$ and RUB-h$^+$-$a$ denote hole transport along a crystallographic direction indicated. For comparison, 1D charge transport simulations (taken from ref. [8]) are indicated by empty black circles and the values reported along the chain direction. The best linear fits to the data are indicated by dashed gray lines.

plot in Fig. 5a the exciton and charge diffusion constants along specific directions in the crystals as a function of the barrier height for (hypothetical) site-to-site exciton or charge transfer along this direction, $\Delta A^{\ddagger}$. In the two-state harmonic approximation and for vanishing free energy difference between the sites of the crystal, which is typically a good approximation for organic crystals, the latter takes the form[50,51]

$$\Delta A^{\ddagger} = \frac{\lambda}{4} - \langle |H_{kl}|^2 \rangle^{1/2} + \frac{1}{\lambda} \langle |H_{kl}|^2 \rangle \qquad (1)$$

where $H_{kl}$ is the excitonic coupling ($H_{kl} = H_{kl}^{\text{XT}} = V_{kl}^{\text{TrESP}}$) or electronic coupling for charge transfer ($H_{kl} = H_{kl}^{\text{HT}}(H_{kl}^{\text{ET}})$) for hole (electron) transfer), and $\lambda$ is the reorganization energy for exciton or hole (electron) transfer. Large values of $\Delta A^{\ddagger}$ compared to $k_B T$ are characteristic of small polaron hopping and small or vanishing values for delocalized polaronic transport. Note that the barrier height Eq. (1) is only defined for $\langle |H_{kl}|^2 \rangle^{1/2} < \lambda/2$; for

larger coupling values the barrier disappears and becomes a minimum[50,51]. We find that the data for exciton diffusion join the trend for charge diffusion rather well (data in blue and green in Fig. 5a)—for both species the logarithm of the diffusion constant increases approximately linearly with decreasing barrier height, in accord with rate theories. The corresponding average IPR of exciton or charge carrier is shown in Fig. 5b. Again, the data for excitons join the trend for charges except for systems with vanishing barrier height. Clearly, for barriers $> 2k_B T \approx 50$ meV (ANT and a6T) the exciton is almost fully localized (as seen before in Fig. 4a), whereas for barriers $< k_B T$ (PDI, DCVSN5, Y6) delocalization occurs.

According to the data in Fig. 5, excitons tend to have larger barrier heights, more localized wavefunctions, and smaller diffusion constants than charge carriers, at least for the systems investigated here. Taking ANT as an example we found that the exciton is fully localized on a molecule whereas the hole is

delocalized over five molecules on average[49], see Supplementary Fig. 15 for a juxtaposition of the two wavefunctions. The main reason for this is that, for the systems investigated, the internal (or "inner-sphere") reorganization energies of the excitons (related to diagonal exciton-phonon coupling) tend to be surprisingly large. For some of the molecules investigated here, we found that the internal reorganization energy is more than twice as large for exciton transfer than for charge transfer. At the same time, the largest excitonic couplings are not significantly stronger than the largest electronic couplings for charge transfer (except for a6T), resulting in larger barrier heights (see Supplementary Table 6).

The large reorganization energies for exciton compared to charge transfer in these molecules can be understood by looking at the NTOs that contribute the most to the S1 transitions. Inspecting these orbitals (Fig. 1) one can clearly observe that every bonding interaction in the HOMO-like NTO becomes an anti-bonding interaction in the LUMO-like NTO and vice versa. This causes large changes in the bond strengths and the bond lengths (up to 0.04 Å, Supplementary Fig. 7) when the geometry of the excited state is optimized resulting in large reorganization energies. In case of hole transport, an electron is merely removed from the HOMO, which can be interpreted as a change of the bonding and anti-bonding interactions of the HOMO to no interactions. Therefore, the changes in bond lengths upon electron removal are smaller than for electronic excitation, typically half as large (up to 0.02 Å, Supplementary Fig. 14), resulting in smaller reorganization energies.

Besides reorganization energy, there are two other differences that contribute to the smaller exciton versus charge diffusivity, albeit to a lesser extent. For the systems considered in this work, the long-range nature of excitonic couplings tends to further increase the localization of excitonic states at the bottom of the excitonic band when compared to results where only nearest-neighbor couplings are included (see Supplementary Fig. 12), thus further decreasing exciton diffusion. This is consistent with what was recently found by Beljonne et al. in ref. [20] for short polymer fibers of P3HT. However, this effect is rather small for all excitonic systems investigated here. Another interesting difference concerns the root-mean-square fluctuations of excitonic and electronic couplings (see Table 1). The former are typically no larger than 10% of the mean values (except for ANT), whereas the latter are typically 50% of the mean value or more[49]. The reason is that the excitonic couplings are only relatively weakly dependent on changes in geometry induced by lattice vibrations due to their Coulombic/long-range nature ($\propto r^{-3}$) compared to the exponential sensitivity of electronic couplings. In the hopping regime (large barriers) coupling fluctuations are beneficial for the localized exciton to overcome the activation barrier and this could be the reason why charge transport in pMSB is somewhat faster than exciton transport in a6T although the two systems feature about the same barrier height and similar crystal packing arrangement.

The most diffusive excitonic systems investigated here, DCVSN5 and Y6, feature essentially barrier-less diffusion owing to large excitonic couplings along the $P_a$ stacking directions (see Fig. 1). Nonetheless, for DCVSN5 the diffusion constant in this direction is a factor of about three smaller than for charge transport in perylene crystals where barrier heights are also vanishingly small. Moreover, the exciton in DCVSN5 delocalizes over only about two molecules whereas the excess electron in perylene delocalizes over almost 12 molecules[49]. The reason is that DCVSN5 forms a stacked structure giving rise to highly anisotropic, 1D exciton transport along the stacking direction, whereas perylene forms 2D layers that permit more isotropic, 2D electron transport. If only a 1D chain (along *a*) of the 2D layer of

perylene is simulated the 1D charge diffusion constant and the wavefunction delocalization reduces to values that are very close to the ones for the exciton in DCVSN5 (Fig. 5, black empty circles), showing that the dimensionality has a major impact on delocalization and diffusion of both excitons and charge carriers. The same conclusion holds as well when directly comparing exciton transport in DCVSN5 and Y6. Y6 shows more isotropic couplings (as well as a larger unit-cell area) within its reticular 3D-like structure and thus a higher diffusion constant than DCVSN5. Importantly, we found that, in Y6, when simulating exciton diffusion in a model with reduced dimensionality, specifically a single 1D-pillar, the exciton diffusion constant along the pillar direction decreases by a factor of 4 compared to the diffusion constant along the same direction in the 3D model (Supplementary Fig. 10c). This is because in the 3D structure the average IPR of thermally accessible excitonic band states is markedly higher than in the 1D model making the transient delocalization mechanism more effective (Supplementary Fig. 10d).

The considerations above provide a road map for the design of molecular organic semiconductors with exciton diffusion constants that are beyond today's best materials (i.e., non-fullerene acceptors). We propose three "design rules" of decreasing importance. Rule 1 is to reduce reorganization energy (ideally, to values <150 meV), while retaining high excitonic couplings so that the barrier Eq. (1) disappears. This is the case as soon as $\langle |H_{kl}|^2 \rangle^{1/2} > \lambda/2$, as, e.g., for hole transport in the high mobility organic crystal rubrene which supports extensive charge delocalization and high diffusion constants in the order of 0.5 cm$^2$ s$^{-1}$[49] (see Fig. 5). Notably, lower internal exciton reorganization energies are the more likely the more similar the bonding and anti-bonding interactions in the HOMO and LUMO orbitals. Invariably, the requirement for orthogonality of these two orbitals limits the extent to which this can be achieved. Rule 2 is to increase the isotropy of excitonic coupling strength in the crystal, which will strongly increase the delocalization of thermally accessible excitonic states resulting in the formation of multiple percolation pathways for barrierless diffusion (see Supplementary Fig. 10 and related discussion). Once this is achieved one enters a regime where exciton delocalization and transport is no longer limited by reorganization energy or anisotropy but by the thermal fluctuations of excitonic couplings. Rule 3 is to reduce these fluctuations as much as possible as they contribute to wavefunction localization (note the difference with respect to the hopping regime where fluctuations are beneficial for transport[20,49]). Rule 3 is already better fulfilled by excitons than by charges owing to the smaller root-mean-square fluctuations of excitonic couplings compared to the electronic couplings, as explained above (see Table 1). Each of the three rules aims to increase the delocalization of the excitonic states at the bottom of the excitonic band to which the exciton will relax via internal conversion before the diffusive regime sets in (disregarding here other possible relaxation channels such as charge transfer exciton formation, intersystem crossing or radiative recombination). Favorable sign combinations of excitonic couplings could form a fourth rule (as proposed for charge transport in refs. [52,53]) and this issue, in connection with the formation of H- and J-aggregation and their different spectroscopic signatures[6], deserves further investigation in forthcoming work. Lastly, the effect of static disorder, while not relevant for the single crystals studied here, is likely to become important in thin-film device applications. It would likely lead to the formation of localized trap states that would slow down the exciton transport and decrease the diffusion constant, similar to the situation for charge carriers[47].

In conclusion, we have introduced a highly efficient non-adiabatic molecular dynamics method, termed FE-SH, enabling the simulation of exciton transport in truly nanoscale organic semiconductors, including application-relevant NFA systems. The methodology makes no prior assumptions with regard to the extent of exciton (de)localization, it includes thermal fluctuations of excitonic couplings and site energies beyond the harmonic approximation. FE-SH is computationally fairly inexpensive because electronic structure calculations are not carried out during dynamics; the overhead compared to classical MD is only a factor of 4–5 per MD step (on a single core) for systems of a few hundred molecules. Therefore it is a powerful and practical tool to predict the mechanism and diffusion constants for exciton transport in molecular materials. Extensions of FE-SH are currently underway that will enable us to model the non-adiabatic processes that underpin the working principle of organic solar cells including exciton dissociation to interfacial charge-transfer states, charge separation, and recombination.

## Methods

**Frenkel exciton surface hopping (FE-SH).** As stated in the main text, we consider singlet electronic excitations (XT) of a multichromophoric system. We restrict the treatment to a single excitonic band formed of local molecular excitations and we disregard other excited states such as triplet states or charge transfer states[39,54]. These can in principle be included in an extension of the model. The singlet electronic states of the system are then described in terms of local or quasi-diabatic molecular excitations (Frenkel excitons). We note in passing, that the term quasi-diabatic means in this context that the non-adiabatic coupling between these states is not exactly zero, but negligibly small. The resultant Hamiltonian, often referred to as Frenkel excitonic Hamiltonian, is written as

$$\hat{H}^{XT} = \sum_k^M \epsilon_k^{XT}(t)|\phi_k^{XT}\rangle\langle\phi_k^{XT}| + \sum_k^M \sum_l^M H_{kl}^{XT}(t)|\phi_k^{XT}\rangle\langle\phi_l^{XT}|, \quad (2)$$

where $|\phi_k^{XT}\rangle = |\phi_k^{XT}(\mathbf{R}(t))\rangle$ represents the state with the exciton localized on molecule $k$ and all other molecules $l$ in the ground state, $\epsilon_k^{XT}(\mathbf{R}(t))$ denotes the energy of that state, $H_{kl}^{XT}(\mathbf{R}(t))$ denotes the intermolecular excitonic coupling between states $|\phi_k^{XT}\rangle$ and $|\phi_l^{XT}\rangle$, $M$ the number of molecules, and $\mathbf{R}(t)$ the time-dependent nuclear coordinates. We label these states with a superscript XT to distinguish them from the hole or excess electron states ($\phi$) investigated previously[8]. The eigenstates of this Hamiltonian are the adiabatic excitonic states forming the excitonic band of the system. Writing the excitonic wavefunction as a linear combination of Frenkel excitons,

$$\Psi(t) = \sum_{l=1}^M u_l(t)|\phi_l^{XT}\rangle, \quad (3)$$

and inserting this expression in the time-dependent Schrödinger equation, one obtains the time-evolution of the excitonic wavefunction within the excitonic band:

$$i\hbar\dot{u}_k(t) = \sum_{l=1}^M u_l(t)\left(H_{kl}^{XT}(\mathbf{R}(t)) - i\hbar d_{kl}^{XT}(\mathbf{R}(t))\right), \quad (4)$$

where we have assumed that the quasi-diabatic excitonic states are orthogonal. This equation for exciton propagation is formally equivalent to the one used before for charge propagation in FOB-SH non-adiabatic dynamics[8,9,16]; Frenkel exciton states merely replace hole or excess electron states. The last term on the right-hand side of Eq. (4) includes the non-adiabatic coupling between the quasi-diabatic states, $d_{kl}^{XT} = \langle\phi_k^{XT}|\dot{\phi}_l^{XT}\rangle$, can be assumed to be negligibly small compared to excitonic coupling, $|i\hbar d_{kl}^{XT}| << |H_{kl}^{XT}|$, and is neglected. We present detailed arguments for the validity of this approximation at the end of this section. In accord with Tully's fewest switches surface hopping algorithm, the nuclear degrees of freedom are propagated on one of the potential energy surfaces (PES) obtained by diagonalizing the Hamiltonian Eq. (2) and denoted $E_a$ ("a" for "active surface"). The nuclear motion couples to exciton motion via the dependences of all exciton Hamiltonian matrix elements on $\mathbf{R}(t)$, see Eq. (4), resulting in diagonal and off-diagonal exciton-phonon coupling. The coupling in the reverse direction (also denoted "feedback") from the exciton to the nuclear motion is accounted for by transitions of the nuclear dynamics ("hops") from the PES of the active eigenstate $a$, $E_a$, to the PES of another eigenstate $j$ using Tully's surface hopping probability[55].

Nuclear forces required to propagate the nuclear dynamics are calculated using the Hellmann–Feynman theorem. The force acting on nucleus $I$ on potential energy surface $E_a$ can be expressed as

$$\mathbf{F}_{I,a} = -\nabla_I E_a = -\left[\mathbb{U}^\dagger(\nabla_I \mathbb{H}^{XT})\mathbb{U}\right]_{aa}, \quad (5)$$

where $\nabla_I \equiv \partial/\partial\mathbf{R}_I$, $\mathbb{H}^{XT}$ is the Frenkel Hamiltonian Eq. (2) in matrix representation and $\mathbb{U}$ the unitary matrix diagonalizing $\mathbb{H}^{XT}$. We refer to ref. [16] for an explicit

derivation of Eq. (5). The nuclear derivatives of the diagonal elements, $\nabla_I H_{kk}^{XT} = \nabla_I[\mathbb{H}^{XT}]_{kk}$, are taken as the gradients of the classical force field potential used to calculate the site energies of excitonic state $k$ (see section "site energies" below). The off-diagonal gradients, $\nabla_I H_{kl}^{XT} = \nabla_I[\mathbb{H}^{XT}]_{kl}, k \neq l$, are taken as the gradients of the excitonic couplings evaluated in the TrESP approximation (see section "excitonic couplings" below), $\nabla_I H_{kl}^{XT} = \nabla_I V_{kl}^{TrESP}$, where $V_{kl}^{TrESP}$ is given by Eq. (12) and can be calculated analytically. Notably, this leads to an order of magnitude improvement of the total energy conservation for FE-SH (to $\approx 10^{-9}$ Ha atoms$^{-1}$ ps$^{-1}$) in comparison to FOB-SH where the off-diagonal derivatives have to be calculated numerically[49].

As for charge transport simulation, the surface hopping algorithm needs to be supplemented with a number of important features so the method can be applied to the calculation of transport properties[8,9,49,56]: decoherence correction, trivial crossing detection, elimination of spurious long-range exciton transfer, and adjustment of the velocities in the direction of the non-adiabatic coupling vector in case of a successful surface hop. They are necessary to improve a number of desirable properties including Boltzmann occupation of the excitonic band states in the long time limit, internal consistency between exciton carrier wavefunction and surface populations of the excitonic band states, and convergence of the diffusion with system size and nuclear dynamics time step. We refer to ref. [17,18,45] for a detailed description and discussion of the importance and the physical underpinnings of these additions to the original fewest switches surface hopping method. Thus, Eqs. (2)–(5), Tully's surface hopping probability (see, e.g., ref. [18] for explicit expression) and the above detection and correction schemes define our Frenkel exciton surface hopping method (FE-SH).

**Diffusion tensor and IPR.** Solving Eq. (4), one obtains the excitonic wavefunction as a function of time, $\Psi(t)$. This gives access to key dynamical properties, e.g., the diffusion tensor, the extent of localization or delocalization of the exciton as a function of time, and the mechanism by which the excitonic wavefunction moves within the material. The (second rank) diffusion tensor components, $D_{\alpha\beta}$, can be obtained as the time derivative of the mean squared displacement along the nine Cartesian components (MSD$_{\alpha\beta}$),

$$D_{\alpha\beta} = \frac{1}{2}\lim_{t\to\infty}\frac{dMSD_{\alpha\beta}(t)}{dt} \quad (6)$$

where $\alpha, \beta$ denote the Cartesian coordinates $x, y, z$ and

$$\begin{aligned}MSD_{\alpha\beta}(t) &= \frac{1}{N_{traj}}\sum_{n=1}^{N_{traj}}\langle\Psi_n(t)|(\alpha - \alpha_{0,n})(\beta - \beta_{0,n})|\Psi_n(t)\rangle\\ &\approx \frac{1}{N_{traj}}\sum_{n=1}^{N_{traj}}\sum_{k=1}^M |u_{k,n}(t)|^2(\alpha_{k,n} - \alpha_{0,n})(\beta_{k,n} - \beta_{0,n}).\end{aligned} \quad (7)$$

In Eq. (7), $\Psi_n(t)$ is the time-dependent excitonic wavefunction in trajectory $n$, $\alpha_{0,n}(\beta_{0,n})$ are the initial positions of the excitonic wavefunction, $\alpha_{0,n} = \langle\Psi_n(0)|\alpha|\Psi_n(0)\rangle$, and the square displacements are averaged over $N_{traj}$ FE-SH trajectories. In the second equation, the coordinates of the exciton are discretized and replaced by the center of mass of molecule $k$ in trajectory $n$, $\alpha_{k,n}$, and $\alpha_{0,n} = \sum_{k=1}^M |u_{k,n}(0)|^2\alpha_{k,n}(0)$, where $|u_{k,n}(t)|^2$ is the time dependent excitonic population of site $k$ in trajectory $n$ as obtained by solving Eq. (4). The elements of the diffusion tensor, $D_{\alpha\beta}$, can be used to define the diffusion lengths[4], $L_{\alpha\beta} = \sqrt{2D_{\alpha\beta}\tau}$, where $\tau$ is the exciton lifetime, i.e., the time it takes for the exciton to relax to the ground state. The latter quantity can be obtained from, e.g., photo-luminescence experiments[4].

We use the inverse participation ratio (IPR) as a measure to describe the delocalization of the excitonic wavefunction $\Psi(t)$,

$$IPR(t) = \frac{1}{N_{traj}}\sum_{n=1}^{N_{traj}}\frac{1}{\sum_{k=1}^M |u_{k,n}(t)|^4}. \quad (8)$$

The numerical value of the IPR is about equal to the number of molecules the wavefunction is delocalized over. A similar definition is used to describe the delocalization of the adiabatic states or eigenstates of the Hamiltonian Eq. (2), $\psi_i^{XT}(t)$,

$$IPR_i(t) = \frac{1}{N_{traj}}\sum_{n=1}^{N_{traj}}\frac{1}{\sum_{k=1}^M |U_{ki,n}(t)|^4}, \quad (9)$$

where $U_{ki,n}$ are the components of the eigenvector $i$ of the Hamiltonian Eq. (2) in trajectory $n$.

**Excitation energies.** To establish a suitable level of theory for parametrization of the excitonic Hamiltonian (Eq. (2)), we started by computing the excitation energies of ANT, a6T, PDI, DCVSN5, and Y6 single molecules in a vacuum with two different long range-separated hybrid functionals, CAM-B3LYP[57] and $\omega$B97X-D[58]. These functionals are commonly used for excitation energy calculations on organic molecules due to their correct asymptotic behavior and balanced description of locally excited and charge-transfer states. The basis set was fixed to 6-31G(d,p) as commonly used in the literature for similar systems[29,59,60]. The vertical and adiabatic excitation energies are summarized in Supplementary

Table 1. All quantum chemical calculations have been performed using the TDDFT implementation in the Gaussian16 software package[61].

**Reorganization energies.** Exciton reorganization energies[29,62], $\lambda^{XT}$, are divided into an internal ("inner-sphere") and an external ("outer-sphere") contribution. The internal contribution, corresponding to the first excited S1 state of the system, is calculated in a vacuum using the four-point scheme,

$$\lambda_i^{XT} = [E_{S1}(\mathbf{R}_{S0}) + E_{S0}(\mathbf{R}_{S1})] - [E_{S1}(\mathbf{R}_{S1}) + E_{S0}(\mathbf{R}_{S0})]. \tag{10}$$

where $E_{S0}$ and $E_{S1}$ denote the potential energy of the S0 ground state and S1 excited state, respectively, and $\mathbf{R}_{S0}$ and $\mathbf{R}_{S1}$ are the nuclear coordinates at the minimum of the S0 and S1 potential energy surfaces. The values are reported in Supplementary Table 2. The external reorganization energy was calculated for anthracene using QM/MMpol[63,64]. We found it to be negligibly small: 0.3 meV and neglected this contribution also for the other systems. Thus, we set $\lambda^{XT} = \lambda_i^{XT}$.

**Site energies.** The approach chosen for the calculation of site energies, that is, the diagonal elements of the excitonic Hamiltonian of the aggregate (Eq. (2)), is similar to the approach used for charge transport in our previously developed FOB-SH[17,18]. The potential energy of the excitonic state $\phi_k^{XT}$ is obtained from classical force fields, where molecule $k$ (molecules $l \neq k$) are described with force field parameters parametrized to the singlet excited (ground) state of the molecules as obtained from the previous TDDFT calculations. In particular, the equilibrium distances for bonding interactions in the excited state are displaced with respect to the ones for the ground state to reproduce the exciton reorganization energy, Eq. (10) (see Supplementary Fig. 7).

**Excitonic couplings.** The excitonic coupling between states $|\phi_k^{XT}\rangle$ and $|\phi_l^{XT}\rangle$, $H_{kl}^{XT}$, is computed in the dimer approximation in a vacuum. Molecules $k$ and $l$, with geometries taken from the condensed phase (crystal structure or FE-SH simulation), form the donor-acceptor pair and the two exciton states are denoted $|k^*l\rangle$ and $|kl^*\rangle$, respectively. The excitonic coupling in the dimer approximation is denoted $V_{kl} = \langle k^*l|\hat{H}_{dimer}|kl^*\rangle$. Reference excitonic couplings $V_{kl}$, shown in Fig. 2, were calculated using a multi-state diabatization procedure MS-FED-FCD[42,43,65]. First, the adiabatic excited states of the donor-acceptor pair is calculated followed by diabatization to localized, (maximally) diabatic excitonic states[43]. The excitonic coupling corresponds to the off-diagonal element between the locally excited (Frenkel) states and includes all long- and short-range contributions. We found that a two-state adiabatic basis is not sufficient to retrieve completely localized states: in fact, an adiabatic state could be the combination of many diabatic states of both donor and acceptor. Moreover, charge transfer states can mix with Frenkel exciton states, and vice versa[42,65]. To overcome this difficulty and recover the coupling between completely de-mixed Frenkel exciton states—which form the state space for the Frenkel Hamiltonian in Eq. (2)– we included up to 20 excited states of the donor-acceptor systems to ensure a complete de-mixing between excitations of different nature and an optimal reconstruction of localized Frenkel exciton states and related couplings. We refer to Supplementary Figure 1 and Supplementary Figure 2 and related discussion for a convergence analysis.

As explained in the main text, $V$ can be written in terms of a long-range or Coulomb term and a short-range term, $V_{kl} = V_{kl}^{Coulomb} + V_{kl}^{short}$[40,66]. The Coulomb term is defined by the Coulombic interaction between the transition densities of isolated donor and acceptor (singlet) states,

$$V_{kl}^{Coulomb} = \int d\mathbf{r} \int d\mathbf{r}' \rho_k^{T*}(\mathbf{r}') \frac{1}{|\mathbf{r} - \mathbf{r}'|} \rho_l^{T}(\mathbf{r}), \tag{11}$$

where $\rho_k^T$ and $\rho_l^T$ are the diagonal parts of the one-particle density matrix constructed from the ground and excited-state wave functions of isolated donor and acceptor. The transition densities of each molecule can be computed efficiently through an atomic orbital expansion in combination with configuration interaction singles (CIS) and TDDFT[40]. Note that transition densities are computed on isolated donor/acceptor in these calculations, as opposed to MS-FED-FCD calculations (where a calculation on the whole donor-acceptor pair is performed). The total excitonic coupling, as well as the Coulomb term Eq. (11), were computed within the TDDFT framework implemented in the Gaussian16 software package[61,67] employing the same functionals and basis set as for the calculation of excitation and reorganization energies above.

During FE-SH propagation, the transition densities are approximated by transition charges resulting in the following expression for the Coulomb integral Eq. (11):

$$V_{kl}^{TrESP} = \sum_{A \in k}^{N} \sum_{B \in l}^{N} \frac{q_A^T q_B^T}{|\mathbf{r}_A - \mathbf{r}_B|}, \tag{12}$$

where the indices $A$ and $B$ run over the atoms of molecules $k$ and $l$, respectively, $q_A^T$, $q_B^T$ are the transition charges and $\mathbf{r}_A$, $\mathbf{r}_B$ are the positions of atoms $A$ and $B$, respectively. Herein, we use the transition charges obtained from the fitting of the electrostatic potential (TrESP). TrESP charges are obtained as proposed by Renger et al.[41], fitting the electrostatic potential generated by the transition density. When used in the FE-SH framework, TrESP charges are computed once for the isolated

molecule in a vacuum, prior production run, and then kept frozen along the trajectory (see Table 1, Fig. 2, and Supplementary Note 3 for a discussion on the accuracy of this approach). The error due to keeping the transition charges fixed along dynamics was investigated and found to be rather small (see discussion in Supplementary Note 3 and related Supplementary Figs. 3, 4).

**FE-SH simulation details.** For each system, we built a series of supercells from the experimental crystallographic unit cell. The dimensions of some of the largest supercells generated are summarized in Supplementary Table 4. For ANT, a6T, and PDI, the intra- and inter-molecular interaction terms for the ground state are taken from the generalized amber force field (GAFF)[68]. They describe well the structure and vibrational properties of acenes and conjugated systems[69]. For DCVSN5 and Y6 we employed the parametrization proposed in ref. [29] and ref. [70], respectively. We refer to the latter references for a full description of the parametrization procedure and related validation.

These supercells were equilibrated in periodic boundary conditions[17], with all molecules in the ground state, for at least 250 ps in the NVT ensemble to a target temperature of 300 K using a Nosé-Hoover thermostat. This was followed by equilibration for at least 250 ps in the NVE ensemble. From the NVE trajectory, an uncorrelated set of positions and velocities were chosen as starting configurations for FE-SH simulations. Molecules within a given region of the supercell were treated as electronically active, i.e., as molecular sites or fragments for construction of the Frenkel Hamiltonian Eq. (2), as previously done for charge transport with FOB-SH[8,49]. All other molecules of the supercell were treated electronically inactive and interacted with the active region only via non-bonded interactions. The number of electronically active molecules was ca. 300 for each system simulated as shown in Supplementary Fig. 9. For ANT, a6T, and PDI, we consider active molecules only within the $a - b$ plane, which is the plane where the strongest exciton interactions are present. The DCVSN5 and Y6 crystals form a different crystal structure compared to the aforementioned systems. DCVSN5 arranges in antiparallel $\pi$-stacked columns with a short inter-columnar distance, whereas Y6 forms a porous inter-penetrating supramolecular structure (see Fig. 1). The largest excitonic couplings for both systems are along the $\pi$-stacked pillars formed by the electron-accepting and donating units of neighboring molecules (a crystallographic direction). However, these systems exhibit also inter-columnar excitonic interactions in directions approximately perpendicular to $a$. To account for the inter-columnar interactions we include a number of columnar stacks in our model (5 for DCVSN5 and 4 for Y6) resulting in an electronically active region of a tubular, rod-like shape. Though, the lateral dimension of the tubular stacks is not sufficiently large for the calculation of exciton diffusion constants in that direction. Thus, our simulations only yield the diffusion constant along the $\pi$-stacked pillars in these two materials. We remark that the supercell dimensions for the different systems chosen in this work constitute the best possible compromise between the accuracy of the results and computational cost. Nevertheless, it is worth mentioning again that wavefunction delocalization and diffusion constants as found for the present 2D planes and rod-like nano-crystal models might constitute a lower bound for the corresponding values in the full 3D bulk crystal.

The initial excitonic wavefunction is chosen to be initially localized on a single active molecule $m$, $\Psi(t=0) = \phi_m^{XT}$, where $\phi_m^{XT}$ is the Frenkel exciton located on molecule $m$ and propagated in time according to the FE-SH algorithm described above in the NVE ensemble. After an initial relaxation, the diffusion constant and average IPR are independent of the initial conditions (within the accuracy of our method), as they should be in the diffusive regime. This is verified in Supplementary Fig. 11 where the excitonic wavefunction $\Psi(t)$ of each classical trajectory was started from an eigenstate of the Hamiltonian (at $t = 0$) for a given initial structure sampled as described above. The nuclear time step, $\Delta t$, was chosen as small as 0.01 fs to prevent more trivial crossings (which becomes more problematic for narrow excitonic bandwidth and localized excitonic wavefunction). Eq. (4) was integrated using the fourth-order Runge–Kutta algorithm and an electronic time step of $\delta t = \Delta t/5$. For each system, about 600 FE-SH trajectories of the length of at least 1 ps were run for calculation of the diffusion tensor and IPR. The components of the MSD Eq. (7) were block averaged over three blocks (about 200 trajectories each) for the calculation of error bars (see Supplementary Fig. 8). The convergence of the diffusion constant with the number of electronically active molecules is shown in Supplementary Fig. 9. All simulations were carried out with our in-house implementation of FE-SH in the CP2K simulation package[71].

**Neglect of $d_{12}^{XT}$ in Eq. (4).** We consider a simple two-site model with one electron on each site occupying either the ground state $\chi_k^{H}$ or the excited state $\chi_k^{L}$, $k$ the site index, $k = 1, 2$. In the initial state (1) the electron on site 1 is in the excited state and the electron on site 2 is in the ground state and vice versa for the final state (2). The corresponding wavefunctions are $\phi_1^{XT} = 2^{-1/2} \det(\chi_1^L(r_1)\chi_2^H(r_2))$ and $\phi_2^{XT} = 2^{-1/2} \det(\chi_1^H(r_1)\chi_2^L(r_2))$. It can be shown that the non-adiabatic coupling between the two states, $d_{12}^{XT} = \langle \phi_1^{XT}|\dot{\phi}_2^{XT}\rangle$, is given by $d_{12}^{XT} = -(\langle \chi_1^L|\dot{\chi}_2^L\rangle \langle \chi_2^H|\chi_1^H\rangle + \langle \chi_2^H|\dot{\chi}_1^H\rangle \langle \chi_1^L|\chi_2^L\rangle)$, where we have assumed that the ground and excited states on the same site are orthogonal, $\langle \chi_k^L|\chi_k^H\rangle = 0$. For multi-electron sites with the S1 excitation dominated by the HOMO-LUMO transition, the above expression with L the LUMO orbital and H the HOMO orbital becomes the leading terms of non-adiabatic coupling.

According to this expression, the non-adiabatic coupling is the product of the time-dependent change in LUMO (HOMO) overlap on different molecules $k$ and $l$ times the HOMO (LUMO) overlap, also taken on molecules $k$ and $l$. Each of the two terms are exponentially decaying with distance and are very small at molecular separations compared to the leading (Coulombic) term of excitonic coupling. In fact, the situation for excitons is even more favorable than for hole or excess electron transfer, where it was shown that a similar approximation induces virtually no changes in the electron dynamics[49]. In the latter case, the leading term to non-adiabatic coupling is proportional to the time-dependent change in LUMO (HOMO) overlap, only. For exciton transfer, the additional HOMO (LUMO) overlap term decreases the non-adiabatic coupling between the two quasi-diabatic states further. Thus, we expect this approximation to hold for excitons even better than for hole or excess electron transfer.

## Data availability

The datasets generated during and/or analyzed during the current study are available in the Zenodo repository, https://doi.org/10.5281/zenodo.6406144. The full data for this study totals more than six TB, so it is in cold storage accessible by the corresponding authors and available upon reasonable request.

## Code availability

The custom codes used for this study are available from the corresponding authors upon request.

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

## Acknowledgements

The authors would like to acknowledge Prof. David Beljonne and Prof. Benedetta Mennucci for useful discussions. S.G. would like to thank Dr. Orestis G. Ziogos and Dr. Matthew Ellis for important inputs on the classical force fields and wavefunction visualization tool respectively employed in this work. S.G. and W.-T.P. were supported by the European Research Council (ERC) under the European Union, Horizon 2020 research and innovation program (grant agreement no. 682539/SOFTCHARGE). Via our membership of the UK's HEC Materials Chemistry Consortium, which is funded by EPSRC (EP/L000202, EP/R029431), this work used the ARCHER UK National Super-computing Service (http://www.archer.ac.uk) as well as the UK Materials and Molecular Modeling (MMM) Hub, which is partially funded by EPSRC (EP/P020194), for computational resources. We also acknowledge the use of the UCL Kathleen High-Performance Computing Facility. D.P. acknowledges the Italian Ministry of Education, University, and Research (MIUR) for a Rita Levi Montalcini grant. L.C. acknowledges the ERC under the grant Lifetimes-716714.

## Author contributions

S.G. implemented the FE-SH code, with input from A.C., S.G. also performed most of the quantum chemical calculations, parametrization of exciton electronic Hamiltonian, and FE-SH simulations of exciton dynamics for the systems investigated in this work. W.-T.P. parametrized and contributed simulation of exciton diffusion in Y6, with input from D.P. and S.G. The MS-FED-FCD was developed by L.C., who also helped with exciton coupling calculations and data interpretation. J.B. contributed to the development of the theoretical framework and the data interpretation, and he supervised all aspects of the research. S.G. and J.B. designed the research and wrote the manuscript. All authors reviewed and discussed the manuscript.

## Competing interests

The authors declare no competing interests.
