## [Peer Review File · Nature Communications]

Exciton transport in molecular organic semiconductors
boosted by transient quantum delocalizationReviewers' Comments:

Reviewer #1:

Remarks to the Author:

Giannini et al. report a theoretical study of exciton diffusion in a collection of five representative molecular materials. The work is thorough and well-described, and I do not have many technical concerns with the methodology. The topic is also timely given experimental advances in recent years, which have furnished several high-profile examples of fast exciton diffusion in organic materials with clear technological relevance. My main concerns are: 1) the strength of the authors main conclusion, 2) the novelty of the design rules they present, and 3) the comparison of exciton and charge transport. My view is that this paper may be suitable for publication in Nature Communications, if the authors are able to address these points.

Before commenting on these issues, I would like to make clear that I am impressed with the authors method, which strikes the right balance between accuracy and speed, and is very well benchmarked and justified. However, as the authors acknowledge, it is essentially the same method that they have presented previously (References 17-19). From the point of view of quantum dynamics, exciton and charge diffusion are isomorphic problems, so extending the existing method to model excitons rather than charges is not very surprising. For me, the impact of this work comes down to the physical and materials design insights the authors can achieve, not the method itself.

1) Strength of the Main Conclusion. The authors' central claim is that exciton transport is boosted by transient delocalization, an example of which is shown in Figure 3b, where the exciton is briefly delocalized over ~ 12 molecules. However, from the data presented, it is unclear how frequent these large delocalization events are, and what their contribution to the calculated diffusivity actually is. Put another way, while Figure 3b demonstrates that transient delocalization events occur in DCVSN5, more detailed analysis is required to prove that these events have a meaningful impact on the average exciton diffusion rate (i.e. compared to the majority of other time points in the calculation, which could potentially be interpreted as hopping of excitons occupying ~ 2 molecules on average).

2) Novelty of the Design Rules. In the final discussion, the authors propose 3 design rules for fast exciton diffusion, which is good to see. The problem is that Rules 1 and 3 are obvious from a theoretical standpoint. Reducing the reorganization energy (Rule 1) will increase incoherent diffusion rates according to standard rate theories and enable coherent motion to be achieved at lower coupling strengths. And once coherent motion is achieved, minimizing electronic coupling fluctuations (Rule 3) will clearly help to minimize energetic disorder and preserve band transport. In my view, it would be better to expand the discussion of Rule 2, which is currently quite vague (perhaps illustrating in a semi-quantitative way how coupling isotropy can improve exciton transport). At present, the design rules do not provide much insight for materials development.

3) Comparison of Charge and Exciton Reorganization Energies. I am surprised that the reorganization energies calculated for hole transfer (Table S4) are so much lower than those for energy transfer (by factors of 2-4). Intuitively, I would expect a larger rearrangement from removing an electron from a molecule, than promoting an electron between orbitals. Given that the first half of the Discussion section focusses on the comparison between charge and exciton transport, this is important to get right. To be confident that the calculated values are realistic, I would suggest that the authors check the basis set/functional dependence of the reorganization energies to see if these values are robust. The external reorganization energy (i.e. polarization of the surrounding molecules) is also likely to be larger for charges than for excitons. It would be good for the authors to estimate these effects, perhaps using an electrostatic model such as that employed in PCCP 2019, 21, 25023-25034.

Reviewer #2:

Remarks to the Author:

The transport of excitons in organic semiconductors is important both for practical applications in organic electronics, as well as a fundamental question in the physics of transport. The problem has not been thoroughly addressed so far because of the computational difficulty in combining all the necessary ingredients, including quantum delocalisation, coupling to an environment, and non-adiabatic transitions.

The manuscript develops a new non-adiabatic molecular dynamics method that is able to overcome the most significant computational challenges. The result is one of the most sophisticated theoretical treatments of exciton diffusion proposed to date. The paper is significant for a few reasons:

1. It explains new physics of exciton transport, which the authors call transient delocalisation, and which is analogous to the same process they uncovered in charge transport.
2. It applies the method to a wide range of organic semiconductors, showing remarkable agreement with experiment.
3. The method is a fully atomistic approach that spans from atomistic scale to long-range diffusion, and the authors have carefully benchmarked every ingredient of their calculation and justified their approximations.

Overall, this is an outstanding paper that I have no hesitation for recommending for publication in Nature Communications.

Below I list some suggestions for improvement, none of which bring into question the authors' conclusions.

On p 3, I think it would be good to define "quasi-adiabatic" because it's not a common term and one that sometimes has different meanings in different papers.

On p 6, the authors refer to the "initially fully localized exciton". I would appreciate a justification (or discussion) of this initial state. In particular, I cannot think of a physical process that would create such a state; in particular, optical excitation would populate eigenstates). I would also appreciate a statement as to how the initial site is chosen.

On p 9, the authors discuss the size of the reorganisation energy in their molecules. At first, I found the values of λ presented in Table 1 to be surprisingly large. Although it's possible that my intuition is miscalibrated, a comparison with similar predictions from elsewhere would be reassuring. Furthermore, as the authors point out, the large size of λ is responsible for their finding that the excitonic states in these systems are localised to 1 or 2 molecules. I found this surprising as well, because excitons in a wide variety of molecular crystals are more delocalised than charges. The prototype examples are J aggregates, where IPRs can reach into tens or even hundreds (e.g., <https://www.pnas.org/content/111/33/E3367>), in molecules that are not too different from the ones studied here. So I think the authors should explain why their theory is consistent with the existence of J aggregates.

On p 9, I am surprised by the result that long-range couplings increase localisation at the bottom of the exciton band. I agree that the results in Fig S9 support this conclusion, but I wonder if the authors have a physical explanation of this observation.

On p 10, in the section on design rules (or elsewhere in the discussion), I think the authors should discuss static disorder, which is present in most devices but is not included in their model. Energetic disorder is smaller for excitons than for charges, but do the authors think it could be important in devices?

In various equations on p 13, I would replace the notation $|u_{\{k,n\}}|^2(t)$ with $|u_{\{k,n\}}(t)|^2$.

It's not clear to me how "Excitation energies" (section on p 14) are different from "Site energies" (section on p 15). I appreciate that there is a difference between isolated molecules and molecules along a trajectory, but I think it would make sense to discuss these together.

The caption to Fig. 2 states "In both panels the mean relative unsigned error is ca. 7%." I think it would be good to make this more precise. E.g., does it mean that each data point should be assumed to have $\pm 7\%$ error bars in both directions, or that the average disagreement between the two methods is 7%? I would also encourage the authors to state a take-away conclusion from the plots. E.g., I think the point is that V_{Coulomb} is generally better than V_{TrESP} , but both may be suitable for certain applications.

Fig. 3 is very informative, but it left me wondering where the energy goes (or comes from) in the various delocalisation events. E.g., in ANT, within 100 fs there are three transitions where E_a changes by over 200 meV. E_a is the nuclear potential energy, and it's surprising that the excitonic potential energy also goes up during the transitions, with the a -index jumping considerably as well. Obviously, these are not thermal transitions, so the 200+ meV of energy come from other degrees of freedom, which are not shown. I think it might be helpful to readers to show where the energy comes from that drives these very energetic (relative to kT) transitions.

In Fig. 4a, I would not connect the data points with the thin gray line.

In Table 1:

- a) my feeling is that too many decimal figures are being reported for almost all of the quantities, especially since the σ^{TrESP} column implies that none of these are really meaningful to better than a few meV.
- b) The units of "Dist." should be given.
- c) The second column should have a label, or at least the caption should explain what P_b , T , P_a , etc. are.

In Table 2:

- a) I would be careful with significant figures in the uncertainties. E.g. 4.5 ± 0.29 should probably be 4.5 ± 0.3 .
- b) If there's an uncertainty in D , I would expect a corresponding uncertainty in L .

Reviewer #3:

Remarks to the Author:

Overall, this is a very nice paper. I particularly enjoyed the figures showing the failure of the point dipole approximation for some of these systems (Figure 5 in the SI). The authors have come up with a clever scheme that permits a reasonable exciton model while exploiting the computational simplicity of force field based descriptions of the excitonic state energies and couplings. Their work clarifies excitonic delocalization in molecular crystals and this will be an important contribution to Nature Comm.

The authors are missing references to Acc Chem Res v47 p2857 (2014) and PCCP v19 p14863 (2017), where previous workers describe both excited state dynamics and nonadiabatic dynamics in the context of an ab initio exciton model. The present authors' use of reparameterized empirical force fields for site energies and precomputed (geometry-independent) charges for the determination of excitonic couplings is of course different from the previous work (which computed these terms from first principles at every time step) and this allows them to look at significantly larger systems for a long time. But the basic idea of surface-hopping with an ab initio derived "on the fly" exciton model

was already done for more than 3000 atoms in the PCCP paper and the authors should both acknowledge this and provide the reader with some explicit details about what is new here compared to that work. To be clear, I believe the authors' approximations are all quite reasonable and I very much appreciate their work. But a clear comparison with the PCCP will help readers understand what is being done differently here.

An important question concerns the periodic simulation. Are the excitonic wavefunctions periodically replicated outside the primary cell? Are the authors using a gamma-point approximation? Is the simulation cell large enough to ensure that gamma point is sufficient? Have the authors tested that? Or are they doing periodic MD with the force field and then ignoring the periodicity within the exciton model? If the latter, it would be good to show that the interactions across cells vanish or otherwise justify their approach as an impurity model and explicitly discuss the implications.

Reviewer 1:

Giannini et al. report a theoretical study of exciton diffusion in a collection of five representative molecular materials. The work is thorough and well-described, and I do not have many technical concerns with the methodology. The topic is also timely given experimental advances in recent years, which have furnished several high-profile examples of fast exciton diffusion in organic materials with clear technological relevance. My main concerns are: 1) the strength of the authors main conclusion, 2) the novelty of the design rules they present, and 3) the comparison of exciton and charge transport. My view is that this paper may be suitable for publication in Nature Communications, if the authors are able to address these points.

Before commenting on these issues, I would like to make clear that I am impressed with the authors method, which strikes the right balance between accuracy and speed, and is very well benchmarked and justified. However, as the authors acknowledge, it is essentially the same method that they have presented previously (References 17-19). From the point of view of quantum dynamics, exciton and charge diffusion are isomorphic problems, so extending the existing method to model excitons rather than charges is not very surprising. For me, the impact of this work comes down to the physical and materials design insights the authors can achieve, not the method itself.

We sincerely thank the reviewer for the positive comment. Below we addressed all the points raised. We would like to remark that, although the surface hopping methodology employed here is similar for charges and for excitons as the reviewer correctly pointed out, the Hamiltonian parameters (particularly the off-diagonal interactions) and their nuclear derivatives are different and had to be implemented from scratch in the present work – hence, the method presented in this manuscript is a non-trivial extension of our surface hopping method for charge transfer presented in Refs. 17-19 (now Refs. 16-18).

Q1) Strength of the Main Conclusion. The authors' central claim is that exciton transport is boosted by transient delocalization, an example of which is shown in Figure 3b, where the exciton is briefly delocalized over ~12 molecules. However, from the data presented, it is unclear how frequent these large delocalization events are, and what their contribution to the calculated diffusivity actually is. Put another way, while Figure 3b demonstrates that transient delocalization events occur in DCVSN5, more detailed analysis is required to prove that these events have a meaningful impact on the average exciton diffusion rate (i.e. compared to the majority of other time points in the calculation, which could potentially be interpreted as hopping of excitons occupying ~2 molecules on average).

This is a very valuable point which indeed deserves more details, we thank the reviewer for suggesting this analysis to us. To quantify the frequency of transient delocalization events and their impact on the diffusion constant we have analysed the DCVSN5 non-adiabatic dynamics in much detail. The results of this analysis give further support for the main conclusion and, in fact, the title of our paper.

-We have replaced the original panels b-f of Figure 4 (displaying the MSDs for the different systems) by 3 new panels (4b-d) that show the results of our new analysis. The original panels b-f of Figure 4 were moved to the SI (Supplementary Figure 8). in a new section on page 16 called: “**Mean squared displacements**”.

-The reviewer’s question on the frequency of transient delocalization events is addressed in the last paragraph on page 6 referring to new Figure 4b: “**We define events where $IPR(t) > \langle IPR \rangle + 1$ as “transient delocalization” to distinguish them from nearest neighbour hopping which is characterized by $IPR = \langle IPR \rangle + 1$ at the transition state. These transient delocalization events account for 8.4 % of the full IPR distribution (Figure 4b, section shaded in yellow). Averaging over all trajectories, we find that they occur about every ~ 90 fs and typically last less than 8 fs.**

-The reviewers question on the impact of transient delocalization on the diffusion constant is addressed in a new paragraph on page 8 referring to new Figure 4c-d: “**Impact of transient delocalization on diffusion.** Transient delocalization events, by which the exciton wavefunction delocalizes over several molecules at a time (as seen in Figure 3b), have a strong impact on the diffusion constant. We quantified this effect for DCVSN5 by calculating a modified MSD of the exciton wavefunction, $\Psi(t)$, where all displacements that occur via transient wavefunction expansion beyond a given IPR threshold (IPR_t) are discarded. The MSD obtained for a threshold $IPR_t = \langle IPR \rangle + 1 \approx 3$, which includes nearest neighbour hops of a delocalized polaron only, is shown in Figure 4c (red line). In this case, the slope and therefore the diffusion constant is a factor of 3 smaller than for the full MSD (blue line) that includes transient delocalization events (defined as $IPR > \langle IPR \rangle + 1$, see above) showing that the latter boost diffusion significantly (3-fold). To generalize our results, in Figure 4d we show the percentage contribution of wavefunction delocalization events to D as a function of the IPR_t . We find a steady increase in diffusion constant with IPR_t . Even at $IPR_t = 2\langle IPR \rangle \approx 4$, only about half of the diffusion constant is accounted for. More extended delocalization events with $IPR > 2\langle IPR \rangle$, which happen only about 5 % of the time according to Figure 4b, contribute the remaining 50 % to the diffusion constant. This analysis clearly shows the major impact of transient delocalization events on exciton diffusion.”

Q2) Novelty of the Design Rules. In the final discussion, the authors propose 3 design rules for fast exciton diffusion, which is good to see. The problem is that Rules 1 and 3 are obvious from a theoretical standpoint. Reducing the reorganization energy (Rule 1) will increase incoherent diffusion rates according to standard rate theories and enable coherent motion to be achieved at lower coupling strengths. And once coherent motion is achieved, minimizing electronic coupling fluctuations (Rule 3) will clearly help to minimize energetic disorder and preserve band transport. In my view, it would be better to expand the discussion of Rule 2, which is currently quite vague (perhaps illustrating in a semi-quantitative way how coupling isotropy can improve exciton transport). At present, the design rules do not provide much insight for materials development.

Thank you for this suggestion. We would like to point out that prior to this present study we would not have been able to formulate these design rules in order of importance, because we did not know the numerical values of the parameters and to which extent the thermal fluctuations of these parameters impact the transport scenario. This became only apparent through the present simulations. We agree with the reviewer that *in retrospect* some of the rules appear as “unsurprising” and “logical”. For instance, prior to this study I (and my colleagues) would not have thought it is most important for materials design to bring reorganization down to further improve exciton diffusion. This is different from, e.g. improving charge transport, where reorganization is already low in the currently best materials and further progress needs to focus on making transport more isotropic.

Regarding Rule 2, to test our claim that isotropic exciton transport is more efficient than anisotropic transport, we have performed additional exciton transport simulations in reduced dimensionality models and added the main conclusions of our new results in the main text on page 11: “**Importantly, we found that, in Y6, when simulating exciton diffusion in a model with reduced dimensionality, specifically a 1D-pillar, the exciton diffusion constant along the pillar direction decreases by a factor of 4 compared to diffusion along the same direction in the 3D model (Supplementary Figure 10c). This is because in the 3D structure the average IPR of**

thermally accessible excitonic band states is markedly higher than in the 1D model making the transient delocalization mechanism more effective (Supplementary Figure 10d.)". Details of the additional simulations and further explanations are given in the SI on page 18 where we have added a new section "Impact of isotropy and dimensionality on exciton transport" along with a new Supplementary Figure 10.

Q3) Comparison of Charge and Exciton Reorganization Energies.

Q3.1) I am surprised that the reorganization energies calculated for hole transfer (Table S4) are so much lower than those for energy transfer (by factors of 2-4). Intuitively, I would expect a larger rearrangement from removing an electron from a molecule, than promoting an electron between orbitals. Given that the first half of the Discussion section focusses on the comparison between charge and exciton transport, this is important to get right. To be confident that the calculated values are realistic, I would suggest that the authors check the basis set/functional dependence of the reorganization energies to see if these values are robust.

That reorganization energy for excitons is significantly larger than for charges (usually a factor of ~2), also took us by surprise, at first sight. The first point to notice is that for the systems investigated, the total reorganization energy is dominated by the internal (intramolecular) contribution, not the external contribution (see our answer to Q3.2 below). The internal contribution is determined by the change in intramolecular bonding interactions upon excitation/charge removal. For the systems investigated, the excitation involves almost entirely a HOMO to LUMO transition. As illustrated in Figure 1 of the main text, the bonding interactions in the HOMO become anti-bonding in the LUMO and vice versa. This results in a drastic change in bond lengths (bonds that become antibonding become larger, antibonds that become bonding become shorter) giving rise to large internal reorganization energy. For charge transfer, an electron is fully removed from the HOMO which corresponds to a change from bonding to non-bonding interaction. This causes smaller bond length displacements (about a factor of 2 smaller, see Supplementary Figure 14) giving internal reorganization energy about a factor of 2 smaller than for excitons. This explanation was given in the manuscript on page 10, paragraph 2, and additional discussion is given in the SI (on page 28).

Nonetheless, as suggested by the referee, we checked the robustness of our calculations of internal reorganization energy. We have added to our previous calculations performed with two well-established long-range corrected hybrid functionals (CAM-B3LYP and wB97X) also a third functional: M06-2X. As requested by the reviewer, we also tested the influence of the basis-set. The results were added in a new section on page 3 of the SI "Internal exciton reorganization energies". We found that exciton reorganization energies are quite robust with respect to the choice of the functional and they change very little with increasing basis-set size. They also agree quite well with literature data. In addition, when discussing the difference between charge and exciton reorganization energies on page 28 of the SI, we point out that "We note that, for comparison, hole reorganization energies were computed using B3LYP functional, which is the most common and widely used functional for electron/hole reorganization energies^{3,14,50} because of its good agreement with UPS spectroscopic measurements.^{51,52}". Also in the latter case, we checked that our results are robust with respect to basis-set size.

Q3.2) The external reorganization energy (i.e. polarization of the surrounding molecules) is also likely to be larger for charges than for excitons. It would be good for the authors to estimate these effects, perhaps using an electrostatic model such as that employed in PCCP 2019, 21, 25023-25034.

In Figure 5 all charge transport systems considered (naphthalene, pMSB, anthracene, etc.) are apolar. Troisi et al. have shown that for such systems the external (or outer-sphere) reorganization energy is very small (McMahon, D. P. & Troisi, A. *J. Phys. Chem. Lett.* **1**, 941–946 (2010)), significantly smaller than the inner sphere contribution, and can be neglected to a good approximation. We note that this is in stark contrast to charge transfer/transport in dipolar liquids or proteins where the outer-sphere contribution is dominating. To estimate the magnitude of outer-sphere reorganization energy for exciton transfer, we performed QM/MMpol calculations according to Bondanza, M., Nottoli, M., Cupellini, L., Lipparini, F. & Mennucci, B. *Phys. Chem. Chem. Phys.* **22**, 14433–14448 (2020) and Curutchet, C. et al. *J. Chem. Theory Comput.* **5**, 1838–1848 (2009). We found that for anthracene the external exciton reorganization energy is indeed very small and, by all means, negligible: 0.3 meV. We have included this result in the computational section, the last paragraph on page 15.

Reviewer 2:

The transport of excitons in organic semiconductors is important both for practical applications in organic electronics, as well as a fundamental question in the physics of transport. The problem has not been thoroughly addressed so far because of the computational difficulty in combining all the necessary ingredients, including quantum delocalisation, coupling to an environment, and non-adiabatic transitions.

The manuscript develops a new non-adiabatic molecular dynamics method that is able to overcome the most significant computational challenges. The result is one of the most sophisticated theoretical treatments of exciton diffusion proposed to date. The paper is significant for a few reasons:

- 1. It explains new physics of exciton transport, which the authors call transient delocalisation, and which is analogous to the same process they uncovered in charge transport.*
- 2. It applies the method to a wide range of organic semiconductors, showing remarkable agreement with experiment.*
- 3. The method is a fully atomistic approach that spans from atomistic scale to long-range diffusion, and the authors have carefully benchmarked every ingredient of their calculation and justified their approximations.*

Overall, this is an outstanding paper that I have no hesitation for recommending for publication in Nature Communications.

Below I list some suggestions for improvement, none of which bring into question the authors' conclusions.

We thank the reviewer for recommending publication and for the positive comments.

Q1) On p 3, I think it would be good to define "quasi-diabatic" because it's not a common term and one that sometimes has different meanings in different papers.

We used quasi-diabatic to indicate that in general it is not possible to obtain perfectly diabatic states. We have added the following explanation in "Methods" on page 12: "We note in passing, that the term quasi-diabatic means in this context that the non-adiabatic coupling between these states is not exactly zero."

Q2) On p 6, the authors refer to the "initially fully localized exciton". I would appreciate a justification (or discussion) of this initial state. In particular, I cannot think of a physical process that would create such a state; in particular, optical excitation would populate eigenstates). I would also appreciate a statement as to how the initial site is chosen.

As mentioned on page 6, although the initial relaxation of the exciton (first 200-400 fs) does, of course, depend on the initial condition chosen, the subsequent diffusive regime, from which the diffusion constant is obtained, is independent of the choice of the initial condition. We have verified this for DCVSN5 starting from an eigenstate located within 2KbT from the bottom of the excitonic band of the system, see the new Supplementary Figure 11 in the SI. In the main text on page 7, we have added "[...] This is because FE-SH fulfils detailed balance in the long-time limit to a very good approximation.^{17,45} This essential condition ensures that after initial relaxation, independently on the initial starting point, the populations of the excitonic band states reach the Boltzmann populations at long times. See Methods for details and Supplementary Figure 11 where simulations are initialized from electronic eigenstates." And at the end of page 18 of the Method section we added that: "After an initial relaxation, the diffusion constant and average IPR are independent on the initial conditions (within the accuracy of our method), as they should be in the diffusive regime. This is verified in Supplementary Figure 11 where the excitonic wavefunction $\Psi(t)$ of each classical trajectory was started from an eigenstate of the Hamiltonian (at $t=0$) for a given initial structure sampled as described above." For practical reasons, which we don't further expand on in this reply, it is more practical to initialize the runs from a fully localized wavefunction.

Q3) On p 9, the authors discuss the size of the reorganisation energy in their molecules. At first, I found the values of λ presented in Table 1 to be surprisingly large. Although it's possible that my intuition is miscalibrated, a comparison with similar predictions from elsewhere would be reassuring. Furthermore, as the authors point out, the large size of λ is responsible for their finding that the excitonic states in these systems are localised to 1 or 2 molecules. I found this surprising as well, because excitons in a wide variety of molecular crystals are more delocalised than charges. The prototype examples are J aggregates, where IPRs can reach into tens or even hundreds (e.g., <https://www.pnas.org/content/111/33/E3367>), in molecules that are not too different from the ones studied here. So I think the authors should explain why their theory is consistent with the existence of J aggregates.

Regarding the first point about the exciton reorganization energy, we have now extended our set of calculations in Supplementary Table 2 and 3 on page 3 of the SI (as requested also by Reviewer 1) and added a new section called “**Internal exciton reorganization energies**” in the SI. As requested, we have added in Supplementary Table 2 data taken from the literature performed with a similar level of theory. The agreement between our data and literature data is very good and we consider our results robust in terms of functional choice and basis-set size.

We believe there is no contradiction with the study of Eisele et al. (the paper cited by the reviewer) for several reasons. (i) The supramolecular system modelled by Eisele et al. features very small reorganization energy characterized by a small diagonal disorder (about 30 meV), which is at least 3-4 times smaller than the disorder characterizing our systems. (ii) Eisele et al. neglect off-diagonal disorder which we take into account explicitly through MD. This further contributes to the localization of the states at the bottom of the excitonic band in our systems, albeit to a smaller extent than reorganization energy/diagonal disorder. (iii) The authors do not include decoherence (i.e. the wavefunction remains delocalized) in their modelling, whereas FE-SH does include decoherence (and other important features) to account for wavefunction delocalization / localization as discussed in the second paragraph of page 14 (iv) Eisele et al. looked at an ideal J-aggregate (all molecules have head-to-tail transition dipoles) which give rise to larger delocalization of the states at the bottom of the excitonic band compared to ideal H-aggregates (for the same exciton coupling strength). The systems investigated in our work are a mixture of J and H aggregation (see e.g. Supplementary Figure 6). This may cause a different degree of localization. On page 11 of the main text in the section called “Design rules”, we have commented the fact that more work is necessary (and it is ongoing) to uncover the connection between the morphology, degree of HJ aggregation and delocalization of the states.

Q4) On p 9, I am surprised by the result that long-range couplings increase localisation at the bottom of the exciton band. I agree that the results in Fig S9 support this conclusion, but I wonder if the authors have a physical explanation of this observation.

The fact that for the organic crystals investigated in this work the thermally accessible states become more localized when long-range interactions are included (Supplementary Figure 9 is now 12) is entirely consistent with what happens in *short* polymer chains as found in Ref. Prodhon, S., Giannini, S., Wang, L. & Beljonne, D. *J. Phys. Chem. Lett.* **12**, 8188–8193 (2021), as we mentioned on page 10 (second paragraph) and in the SI on page 22. The physical explanation might lie with the fact that increasing the range of the off-diagonal elements different from zero within the excitonic Hamiltonian matrix, effectively increases the amount of off-diagonal disorder, thereby making the tail states more localized. Nevertheless, this trend inverts for states in the middle of the excitonic band (which are not thermally accessed). It is difficult to find a more conclusive explanation for this behaviour as it might be system dependent and characteristic of the specific Hamiltonian of a given system. In fact, for long polymer chains, long-range interactions remarkably increase the diffusion constant by a few orders of magnitude (Ref. *J. Phys. Chem. Lett.* **12**, 8188–8193 (2021)).

Q5) On p 10, in the section on design rules (or elsewhere in the discussion), I think the authors should discuss static disorder, which is present in most devices but is not included in their model. Energetic disorder is smaller for excitons than for charges, but do the authors think it could be important in devices?

We added on the former page 10 now 12 (first paragraph): “**Lastly, the effect of static disorder, while not relevant for the single crystals studied here, is likely to become important in thin-film device applications. It would likely lead to the formation of localized trap states that would slow down the exciton transport and decrease the diffusion constant, similar to the situation for charge carriers.⁴⁷**” Also, when comparing computed with experimental diffusion coefficients for Y6, we state on page 8: “**The static disorder present in the thin-films could explain the somewhat smaller diffusion constant in the experiment.**”

Q6) In various equations on p 13, I would replace the notation $|u_{\{k,n\}}|^2(t)$ with $|u_{\{k,n\}}(t)|^2$.

We have now corrected this in all relevant equations on pages 12 and 13 (corrections were not highlighted in this case)

Q7) It's not clear to me how "Excitation energies" (section on p 14) are different from "Site energies" (section on p 15). I appreciate that there is a difference between isolated molecules and molecules along a trajectory, but I think it would make sense to discuss these together.

We thank the reviewer for pointing this out. We agree that it makes sense to discuss the two quantities close by together. We have re-arranged the paragraph called “**Site energies**” and now moved it just below a modified paragraph called “**Excitation and reorganization energies**” on page 15. In this modified section we clarified the fact that excitation energies refer to first-principles calculations in vacuum. These are necessary to parametrize the FE-SH Hamiltonian. On page 15 we added, “**To establish a suitable level of theory to parametrized the excitonic Hamiltonian [...] in vacuum [...]**”. While site energies are the diagonal elements of the Hamiltonian of the full aggregate, which is evaluated using parametrized force-fields in FE-SH and updated on-the-fly along molecular dynamics as the reviewer pointed out.

Q8) The caption to Fig. 2 states "In both panels the mean relative unsigned error is ca. 7%." I think it would be good to make this more precise. E.g., does it mean that each data point should be assumed to have $\pm 7\%$ error bars in both directions, or that the average disagreement between the two methods is 7%? I would also encourage the authors to state a take-away conclusion from the plots. E.g., I think the point is that V_{Coulomb} is generally better than V_{TrESP} , but both may be suitable for certain applications.

We have now defined the mean relative unsigned error in Fig. 2 caption for clarity: “**Mean relative unsigned error (%) is defined as $MRUE = (\sum_n (|y_{\text{calc}} - y_{\text{ref}}| / y_{\text{ref}})) / n$.**” As requested by the reviewer we have also added a take-home message in the caption of Fig. 2 by writing: “**This small deviation found for the systems investigated justifies the use of V_{TrESP} in FE-SH simulations.**”

Q9) Fig. 3 is very informative, but it left me wondering where the energy goes (or comes from) in the various delocalisation events. E.g., in ANT, within 100 fs there are three transitions where E_a changes by over 200 meV. E_a is the nuclear potential energy, and it's surprising that the excitonic potential energy also goes up during the transitions, with the a -index jumping considerably as well. Obviously, these are not thermal transitions, so the 200+ meV of energy come from other degrees of freedom, which are not shown. I think it might be helpful to readers to show where the energy comes from that drives these very energetic (relative to kT) transitions.

At first, we would like to emphasize that our methodology fulfils both total energy conservation when run in the NVE ensemble and detailed balance (as reported on page 7 and related references), please see also answer to Q2. We have now added a comment in this regard in Methods on page 14: “**Notably, this leads to one order of magnitude improvement of the total energy conservation for FE-SH (to $\sim 10^{-9}$ Ha atoms⁻¹ ps⁻¹) in comparison to FOB-SH where the off-diagonal derivatives have to be calculated numerically.⁴⁹**” So, indeed (as the reviewer already mentioned) during the electronic transitions, i.e. surface hops (incl the three transitions the reviewer mentioned), the energy just flows from kinetic to potential energy and the total energy remains preserved. In accord with Tully’s surface hopping algorithm, a transition/hop between two potential energy surfaces is only possible if their energy difference is equal or smaller than the kinetic energy of the nuclei projected onto the non-adiabatic coupling vector that couples the two electronic potential energy surfaces. We have previously proven that on average the kinetic energy along the non-adiabatic coupling vector is $k_B T / 2$ (13 meV at 300 K), see Carof, A., Giannini, S. & Blumberger, J. *J. Chem. Phys.* **147**, 214113 (2017). Thus one can expect to observe thermal excitations of a couple $k_B T$ (~ 100 meV) after initial relaxation (> 200 - 400 fs), which is indeed what is observed. These transitions need to occur to preserve detailed balance in the long time limit. The somewhat larger excitation energies of 200 meV, the reviewer referred to, occur during the initial relaxation period (< 200 fs) where the system initially relaxes from a high-lying electronic state.

Q10) In Fig. 4a, I would not connect the data points with the thin gray line.

This has been changed as suggested.

Q11) In Table 1:

- a) my feeling is that too many decimal figures are being reported for almost all of the quantities, especially since the σ^{\wedge} TrESP column implies that none of these are really meaningful to better than a few meV.*
- b) The units of "Dist." should be given.*
- c) The second column should have a label, or at least the caption should explain what P_b , T , P_a , etc. are.*

We have reduced the number of figures as requested. We have given the units of “Dist.” In a new footnote and we have explained that “[...] **The subscripts approximately indicate the crystallographic direction along which dimers are oriented.**”

Q12) In Table 2:

- a) I would be careful with significant figures in the uncertainties. E.g. 4.5 ± 0.29 should probably be 4.5 ± 0.3 .
b) If there's an uncertainty in D , I would expect a corresponding uncertainty in L .

Thank you for this comment. We have fixed the number of significant digits in Table 2 and also provided uncertainties for L as suggested.

Reviewer 3:

Overall, this is a very nice paper. I particularly enjoyed the figures showing the failure of the point dipole approximation for some of these systems (Figure 5 in the SI). The authors have come up with a clever scheme that permits a reasonable exciton model while exploiting the computational simplicity of force field based descriptions of the excitonic state energies and couplings. Their work clarifies excitonic delocalization in molecular crystals and this will be an important contribution to Nature Comm.

We thank the reviewer very much for their appreciation.

Q1) The authors are missing references to Acc Chem Res v47 p2857 (2014) and PCCP v19 p14863 (2017), where previous workers describe both excited state dynamics and nonadiabatic dynamics in the context of an ab initio exciton model. The present authors' use of reparameterized empirical force fields for site energies and precomputed (geometry-independent) charges for the determination of excitonic couplings is of course different from the previous work (which computed these terms from first principles at every time step) and this allows them to look at significantly larger systems for a long time. But the basic idea of surface-hopping with an ab initio derived "on the fly" exciton model was already done for more than 3000 atoms in the PCCP paper and the authors should both acknowledge this and provide the reader with some explicit details about what is new here compared to that work. To be clear, I believe the authors' approximations are all quite reasonable and I very much appreciate their work. But a clear comparison with the PCCP will help readers understand what is being done differently here.

Not citing these relevant papers was clearly an omission on our side. They are now cited on page 3. We have also added on page 3 *"In this respect, our method differs from the approach by Sisto *et al.*, where surface hopping coupled with an ab-initio derived "on the fly" exciton Hamiltonian was developed to simulate exciton transfer along a few tens of chromophores (~3000 atoms). FE-SH allows us to carry out non-adiabatic MD simulations of exciton transport on yet larger, truly nanoscale systems (> 10 nm, $\sim 10^3$ molecules, $\sim 10^5$ atoms) on the 1-10 ps time scale."* Our algorithm also implements velocity rescaling, decoherence, trivial crossing detection and elimination of spurious long-range energy transport. We added on page 14 after discussing a few of these algorithms that: *"They are necessary to improve a number of desirable properties including Boltzmann occupation of the excitonic band states in the long time limit, internal consistency between exciton carrier wavefunction and surface populations of the excitonic band states, and convergence of the diffusion with system size and nuclear dynamics time step. We refer to Ref.^{17,18,45} for a detailed description and discussion of the importance and the physical underpinnings of these additions to the original fewest switches surface hopping method."* Without these algorithms, even the most accurate underlying electronic structure method could yield incorrect dynamics for large systems.

Q2) An important question concerns the periodic simulation. Are the excitonic wavefunctions periodically replicated outside the primary cell? Are the authors using a gamma-point approximation? Is the simulation cell large enough to ensure that gamma point is sufficient? Have the authors tested that? Or are they doing periodic MD with the force field and then ignoring the periodicity within the exciton model? If the latter, it would be good to show that the interactions across cells vanish or otherwise justify their approach as an impurity model and explicitly discuss the implications.

The excitonic wavefunction is non-periodic as described on page 13 and written as a linear combination of Frenkel excitons localized on "active" fragments (which form the site-basis in which the Hamiltonian in Eq. 2 is represented). We employ very large super-cells comprised of one hundred to a few hundred unit cells (dimensions given in Supplementary Table 4 of the SI) and activate a subset of more than 300 molecules. We need to choose such large supercells to be able to converge the mean square displacement and diffusion constant of the exciton (see Supplementary Figure 9). Evidently, for such large supercells, the gamma-point approximation is valid. The molecular dynamics is done using periodic boundaries applied to the supercells, as the reviewer correctly assumed (the computational protocol and simulation details are given on page 18 of

the main text). With regard to interactions with neighbouring cells, the excitonic interactions decay to virtually zero beyond 2-3 nm depending on the system (see Supplementary Figure 5) and the supercells used in this work exceed in most of the cases 10 nm along the transport direction. Moreover, the charge is delocalized over no more than 15-20 molecules during transient delocalization events. So we expect interactions across cells to be negligibly small. For the supercells used we also investigated the convergence of the diffusion coefficient with system size showing that the latter is relatively well converged (Supplementary Figure 9).

Additional changes

To keep the main text brief and below the 6000 words limit, we moved the comparison between FE-SH and the Ehrenfest dynamics originally on page 8 of the main text, to page 23 of the SI in Section: “**Comparison between diffusion from FE-SH and the literature**”

In the Supplementary Information, there are a few minor changes in the wording of some sentences that we have highlighted in red. The order of a number of sections was also changed and adapted according to some of the questions from reviewers and also for improved clarity. In particular, we rearranged Section: “Comparison of FE-SH diffusion constants with values from the computational and experimental literature” and discuss each of the systems studied in turn.

Subheading in the Discussion section were removed, and “Code availability” and “Data availability” statements were added in accordance with “formatting instructions”.

We hope that all comments by the referees have been addressed duly and adequately and that the revised version of this paper can now be accepted for publication in *Nature Communications*.

Reviewers' Comments:

Reviewer #1:

Remarks to the Author:

After considering the authors responses and the revised version of their manuscript, I am happy to recommend publication of this study in Nature Communications.

The authors have comprehensively addressed my earlier comments. In particular, the new analysis in Figure 4 – where the contribution of delocalisation events to the diffusion coefficient – is great. I think this is an insightful paper that will be well received.

Reviewer #2:

Remarks to the Author:

In my view, the authors have addressed my comments (and those of the other reviewers) well, and I still think the paper should be published, for the reasons I gave in the first round.

I have two points that I would encourage the authors to consider if they are given an opportunity to submit a final revised version. These are optional and I do not need to see the manuscript again.

1. Both Reviewer 1 and myself (and, according to the authors' letter, themselves) were surprised by the large size of the reorganisation energies reported. I do not dispute the accuracy of the calculation or the correctness of the authors' explanation. I would, however, suggest a few editorial changes to benefit other readers who might be likewise surprised. In particular, I would acknowledge that the results are surprising and the explanation offered quite system-specific. At the moment, a fair bit of the text suggests the results are general. E.g., on p. 10: "The main reason for this is that the internal (or "inner-sphere") reorganization energy (related to diagonal electron-phonon coupling) tends to be significantly larger, typically more than twice as large, for exciton transfer than for charge transfer." There is a caveat at the beginning of the paragraph "at least for systems investigated here", but the sentence on p. 10 is written quite categorically and I think is not true in general. Similarly, in the following paragraph, the opening "The large reorganization energies for exciton compared to charge transfer can be understood by looking at the NTOs that contribute most to the S1 transitions." should probably be qualified with "in these molecules", with an explicit acknowledgment of the specificity of the explanation that is given. This is not a weakness of the manuscript, and the authors could turn it into another "design rule": if the changed HOMO/LUMO bonding in these materials is bad for the reorganisation energy, then a design principle is to design molecules where the HOMOs and LUMOs have similar bonding patterns.

2. I think the authors' answer to question 1 of Reviewer 1 introduced additional, important material that better explains the nature of transient delocalisation. I have a few remarks on this new material:

- a) I do not think that the authors get a fair comparison of what happens without transient delocalisation by discarding long-range events. I think that setting these rates to zero artificially lowers the red line in Fig. 4c and that a fairer comparison would be to replace them with typical short-range events, not zero.
- b) I do not understand the claim on p 6 that, at the transition state, $IPR = \langle IPR \rangle + 1$ and would appreciate a justification.
- c) On p 8 and in Fig. 4d, the authors talk about "percentage contribution" of different events to the diffusion coefficient. I think that the authors should precisely define how the contributions of different events are partitioned from the total D , because it's not obvious that they are additive.

Reviewer #3:

Remarks to the Author:

The authors have addressed my comments satisfactorily.

Reviewer 1:

After considering the authors responses and the revised version of their manuscript, I am happy to recommend publication of this study in Nature Communications.

The authors have comprehensively addressed my earlier comments. In particular, the new analysis in Figure 4 – where the contribution of delocalisation events to the diffusion coefficient – is great. I think this is an insightful paper that will be well received.

We thank the reviewer for recommending publication and for the positive feedback about Fig. 4.

Reviewer 2:

In my view, the authors have addressed my comments (and those of the other reviewers) well, and I still think the paper should be published, for the reasons I gave in the first round.

We thank once more the reviewer for recommending publication.

I have two points that I would encourage the authors to consider if they are given an opportunity to submit a final revised version. These are optional and I do not need to see the manuscript again.

We have now considered in turn the new suggestions.

Q1). Both Reviewer 1 and myself (and, according to the authors' letter, themselves) were surprised by the large size of the reorganisation energies reported. I do not dispute the accuracy of the calculation or the correctness of the authors' explanation. I would, however, suggest a few editorial changes to benefit other readers who might be likewise surprised. In particular, I would acknowledge that the results are surprising and the explanation offered quite system-specific. At the moment, a fair bit of the text suggests the results are general. E.g., on p. 10: "The main reason for this is that the internal (or "inner-sphere") reorganization energy (related to diagonal electron-phonon coupling) tends to be significantly larger, typically more than twice as large, for exciton transfer than for charge transfer." There is a caveat at the beginning of the paragraph "at least for systems investigated here", but the sentence on p. 10 is written quite categorically and I think is not true in general.

Following the reviewer suggestion, we have now made our findings more specific to the molecules considered in this work. On page 10, we have added: “The main reason for this is that, **for the system investigated**, [...],” and, as requested, we have also stressed the fact that reorganization energies of the excitons tend to be “**surprisingly large**” in the same paragraph.

Similarly, in the following paragraph, the opening "The large reorganization energies for exciton compared to charge transfer can be understood by looking at the NTOs that contribute most to the S1 transitions." should probably be qualified with "in these molecules", with an explicit acknowledgment of the specificity of the explanation that is given.

We have now added “**in these molecules**” to the sentence proposed by the reviewer.

This is not a weakness of the manuscript, and the authors could turn it into another "design rule": if the changed HOMO/LUMO bonding in these materials is bad for the reorganisation energy, then a design principle is to design molecules where the HOMOs and LUMOs have similar bonding patterns.

We thank the reviewer for the suggestion, we have now added on page 11 after discussing rule 1 that “**Notably, lower internal exciton reorganization energies are the more likely the more similar the bonding and anti-bonding interactions in the HOMO and LUMO orbitals. Invariably, the requirement for orthogonality of these two orbitals, limits the extent to which this can be achieved.**”

Q2). I think the authors' answer to question 1 of Reviewer 1 introduced additional, important material that better explains the nature of transient delocalisation. I have a few remarks on this new material:

We address the comments in turn.

a) I do not think that the authors get a fair comparison of what happens without transient delocalisation by discarding long-range events. I think that setting these rates to zero artificially lowers the red line in Fig. 4c and that a fairer comparison would be to replace them with typical short-range events, not zero.

We would like to point out that, in fact, the impact of transient delocalization can be quantified in different ways. Here we are interested in understanding the *contributions* of true-short range and true long-range transfer events to the diffusion constant. What the reviewer suggests is to replace all physical long-range transfer events with artificial short-range hopping. This would be certainly possible, but in our opinion a decomposition as done in the present paper has more physical relevance.

b) I do not understand the claim on p 6 that, at the transition state, $IPR = + 1$ and would appreciate a justification.

We made the point about $\langle IPR \rangle + 1$ more clear by writing on page 6 that. "We define events where $IPR(t) > \langle IPR \rangle + 1$ as “transient delocalization” to distinguish them **from local or short-range exciton transfer events where, during the transition, $IPR = \langle IPR \rangle + 1$, but not larger than that. This would describe, for instance, hopping of a fully localized exciton to one of its nearest neighbours, where IPR changes from 1 to 2 in the transition state and back to 1 after the transition, or a shift of a delocalized polaron by one molecular unit.**”

c) On p 8 and in Fig. 4d, the authors talk about "percentage contribution" of different events to the diffusion coefficient. I think that the authors should precisely define how the contributions of different events are partitioned from the total D, because it's not obvious that they are additive.

We agree with the reviewer. We have now specified in the y-axis of Fig. 4d that we plot: “ **$(D_{thr} / D) \times 100$** ”. Moreover to make the definition to D more precise, on page 8 we now have replace a sentence with “**To generalize our results, in Figure 4d we show the diffusion constant, D_{thr} , corresponding to an IPR threshold, IPR_{thr} , as a fraction of the total diffusion constant D. In the limit of large IPR_{thr} , i.e., all transitions included irrespective of extent of delocalization, $D_{thr} \rightarrow D$** ”. To avoid confusion between $IPR(t)$ (as a function of time) and the previous IPR_t (for the threshold), IPR_t has been replaced with IPR_{thr} in all occurrences.

Reviewer 3:

The authors have addressed my comments satisfactorily.

We thank the reviewer once more for their appreciation.

Additional changes

All the required changes in the Author Checklist provided by the editor have been made in this revised version of the manuscript and addressed within the Author Checklist file.

We hope that all comments by the referees have been addressed duly and adequately and that the revised version of this paper can now be accepted for publication in *Nature Communications*.